# Compositional Multimodal Reasoning for Long-Horizon Robotic Manipulation in Scientific Experiments

## Abstract

Long-horizon robotic manipulation in scientific experiments requires strict procedural dependencies, multi-stage reasoning, and domain-aware manipulation skills that remain challenging for existing multimodal planning systems. Existing Vision-Language-Action (VLA) models excel at multimodal understanding but often lack explicit symbolic knowledge, limiting their compositional and interpretable planning ability. We present Compositional Multimodal Planner (CoMP), a hierarchical reasoning framework that decouples task understanding, perceptual reasoning, and skill execution for complex experimental procedures. CoMP consists of: (1) a task-level interpreter using chain-of-thought prompting to infer task logic, (2) a mid-level multimodal planner that integrates future scene prediction to enable visually grounded reasoning, and (3) a low-level skill controller that executes actions via reinforcement learning. This decoupled design enables each component to be optimized independently, improving controllability, extensibility, and generalization without fine-tuning large models. To facilitate evaluation, we introduce a benchmark dataset for scientific experiment tasks. Experiments on both our benchmark and RLBench show that CoMP achieves strong performance and superior compositional generalization compared to competitive baselines, highlighting the advantages of structured and decoupled multimodal planning for long-horizon scientific workflows.

## 1 Introduction

Artificial intelligence is transforming scientific research by enabling robots to conduct complex experimental procedures with minimal human intervention, such as high-throughput material synthesis or standardized testing. While early robotic systems (Pyzer-Knapp et al., 2022; Wang & Zhu, 2024; Ramos et al., 2025) have achieved reliable automation in structured environments, real-world scientific tasks are often exploratory and under-specified. Scientists typically provide high-level goals that require contextual interpretation, compositional reasoning over procedural steps, and fine-grained object manipulation. These characteristics make long-horizon planning in laboratory settings very challenging.

Recent advances in large language models (LLMs) have demonstrated strong capabilities in task decomposition and logical reasoning (Ha et al., 2023; Dalal et al., 2024; Zhou et al., 2024; Guo et al., 2024), while vision-language-action (VLA) models (Ma et al., 2024; Black et al., 2025; Zhao et al., 2025) attempt to unify perception and reasoning in an end-to-end manner. Despite their success on household and industrial benchmarks, these methods might face limitations in scientific domains. End-to-end VLAs struggle with systematic generalization: as the number of possible compositions of actions, tools, and procedural constraints grows combinatorially, the amount of data required for reliable learning becomes prohibitive. This observation aligns with recent findings in systematic generalization (Lake & Baroni, 2018; Bahdanau et al., 2019), where modularity is key to scaling beyond training distributions. Recent studies further reveal complementary limitations. For example, task-specific fine-tuning of VLAs has been shown to bias models toward superficial goal completion while neglecting the underlying states and dynamics (Li et al., 2024). Such evidence reveals the inherent shortcomings of purely end-to-end learning.

Figure 1: Overview of the CoMP framework. CoMP decomposes long-horizon tasks into modular stages. First, an LLM generates high-level subtasks from a task prompt. A multimodal model then grounds each subtask by conditioning on the current and predicted goal images, producing sequences of action primitives. Finally, a low-level controller executes these primitives in the environment, enabling safe and consistent task completion.

We argue that addressing long-horizon planning in scientific experiments requires compositional multimodal reasoning rather than scaling end-to-end policies. From an information-theoretic perspective, modular architectures reduce the mutual information each component must encode, effectively disentangling representations of task logic, perceptual dynamics, and control skills. This enables data-efficient learning and more robust adaptation to novel task compositions.

To this end, we propose a modular framework, named Compositional Multimodal Planner (CoMP), that explicitly disentangles semantic task reasoning, visual grounding, and low-level control. CoMP integrates three heterogeneous components: a task-level planner, a mid-level multimodal planner, and a low-level skill controller. In scientific experimentation, high-level procedural logic is largely independent of instant visual observations. Thus, the task-level planner focuses on domain knowledge and chain-of-thought (CoT) (Wei et al., 2022) reasoning to generate abstract sub-tasks that correspond to fundamental experimental operations. These sub-tasks must then be grounded to real objects, physical relations, and scene dynamics, which is the main source of transfer difficulty. To handle this, the mid-level multimodal planner combines visual prediction with language-based reasoning. It learns to interpret out-of-distribution objects and future scene changes, and converts perceptual information into semantic action primitives. These action primitives are further executed by a low-level controller that maps them into feasible motion trajectories through reinforcement learning. This modular design improves transfer efficiency, increases interpretability, and reduces training complexity, while allowing each component to specialize in its own function.

To support systematic evaluation, we introduce a new benchmark for robotic scientific experimentation. Unlike existing datasets focused on spatially grounded manipulation, our benchmark emphasizes procedural dependencies, object-centric constraints, and domain-specific correctness. While some robotic systems for long-horizon decision-making in experimental settings have been developed (Andrychowicz et al., 2020; Schmalstieg et al., 2022; Zhu et al., 2021), these methods often focus on specific lab environments. In contrast, we target generalizable experimental workflows, aiming to enable the robots to autonomously adapt to diverse procedural tasks, which is a critical step toward embodied intelligence in scientific domains. To address these limitations, we develop a scientific experiment simulation suite built on the CoppeliaSim (VREP) simulation software (Rohmer et al., 2013). Based on that, we provide a reliable benchmark dataset for experiment tasks. The contributions of this work are as follows:

- We introduce CoMP, a framework that enables heterogeneous multimodal collaboration across language, vision, and skills to address the combinatorial complexity of long-horizon robotic planning in scientific laboratories.

- We introduce a new benchmark dataset for long-horizon experiment tasks. The dataset captures typical characteristics such as procedural dependencies and object-centric manipulation, and is designed to generalize across diverse scientific domains.

- Extensive experiments on both the proposed dataset and a public benchmark show that CoMP outperforms several baselines in terms of task success, procedural correctness, and generalization.

## 2 RELATED WORK

**Robot manipulation learning**   Robotic manipulation has traditionally relied on symbolic planning and control methods such as trajectory optimization (Parr & Russell, 1997; Kaelbling & Lozano-Pérez, 2011), which work well in structured settings but lack flexibility and generalization. Recent advances in LLMs (Radford, 2018; Touvron et al., 2023) have inspired their use in robotic skill learning (Ichter et al., 2022), programmatic task decomposition (Singh et al., 2023), and in combination with RL for low-level control (Dalal et al., 2024), though these approaches often suffer from unstable execution and poor systematic generalization. VLA models aim to unify perception, reasoning, and control (Yu et al., 2025; Sautenkov et al., 2025; Zhang et al., 2025), but purely end-to-end training struggles to scale as task compositions grow combinatorially. CoT-VLA (Zhao et al., 2025), Hi Robot (Shi et al., 2025), and HAMSTER (Li et al., 2025) move toward hierarchical structures by incorporating CoT reasoning or coarse-to-fine path planning, yet they remain tightly coupled through supervised trajectory learning, limiting modular reusability. In contrast, our proposed CoMP explicitly disentangles task decomposition, multimodal prediction, and low-level control into heterogeneous modules. This modularity allows independent optimization without trajectory-level supervision and enables compositional recombination, improving sample efficiency and robustness under domain shifts.

**Autonomous experimental systems for science**   Prior works have focused on autonomous scientific discovery by combining high-throughput experimentation and AI-driven planning. For example, Zhao et al. (2021) leverage automated platforms to explore the stability of perovskites under varying conditions. Coley et al. (2019) develop an organic synthesis platform using robotic arms and modular components to automate complex chemical processes. Burger et al. (2020) propose a mobile chemical robot that performs hundreds of experiments autonomously to discover a novel hydrogen-production catalyst. While these approaches aim at full automation for materials discovery or synthesis, our work focuses instead on experimental assistance during the R&D process to help researchers perform tedious lab tasks more efficiently.

## 3 METHOD

### 3.1 PIPELINE OVERVIEW

Long-horizon robotic planning requires reasoning, visual grounding, and reliable skill execution, which are challenging to achieve within a single end-to-end model. To address this, we introduce a compositional multimodal planner (CoMP) that disentangles these stages (Fig. 1).

CoMP first decomposes high-level instructions into symbolic subtasks using an LLM. Each subtask is then grounded into a sequence of action primitives by a multimodal planner conditioned on the current and predicted goal images. Finally, a low-level controller executes these primitives in the environment, ensuring safe and consistent task completion.

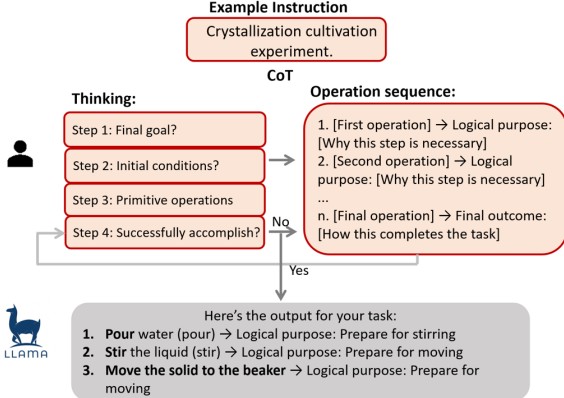

Figure 2: An example of the task-level planner using CoT reasoning. The LLM is guided through a four-step process: (1) clarify the goal, (2) identify required objects and conditions, (3) decompose into basic operations, and (4) validate task completion.

## 3.2 TASK-LEVEL PLANNER

We model the task-level planner as a symbolic decomposition process, operating entirely in the language space and decoupled from perception or control signals. As shown in Fig. 2, given a high-level goal $G$ (e.g., "crystallization cultivation experiment. "), we use CoT reasoning with the Meta-Llama-3.1-8B-Instruct model to generate a sequence of semantically coherent subtasks

$$\mathcal{S} = \{s_1, s_2, \ldots, s_T\}, \quad s_t \in \mathcal{L},$$

where $\mathcal{L}$ denotes the symbolic instruction space (natural-language actions such as "add reagent A").

To construct $\mathcal{S}$, we employ CoT prompting, which decomposes reasoning into four stages:

$$G \xrightarrow{\text{(1) interpret}} g \xrightarrow{\text{(2) prerequisites}} \mathcal{P} \xrightarrow{\text{(3) decompose}} \mathcal{S} \xrightarrow{\text{(4) validate}} \mathcal{S}^*,$$

where $g$ denotes the interpreted core objective, $\mathcal{P}$ is the set of prerequisite conditions, and $\mathcal{S}^*$ is the optimized subtask sequence after logical validation.

To improve reliability, we introduce a verification–correction loop applied iteratively:

$$\mathcal{S}^{(k+1)} = \Phi\big(\mathcal{S}^{(k)}\big), \quad \Phi = f_{\text{val}} \circ f_{\text{cons}} \circ f_{\text{corr}},$$

where $f_{\text{val}}$ performs subgoal validation, $f_{\text{cons}}$ checks temporal/causal/physical constraints, and $f_{\text{corr}}$ applies corrections (see detailed description in Appendix B). This process continues until $\mathcal{S}^{(k)}$ converges to a logically consistent plan $\mathcal{S}^*$. This mechanism operates entirely within the LLM without external labels, leveraging its inherent reasoning abilities to ensure semantic correctness. As task-level planner is performed in the symbolic space of language, it is decoupled from direct visual inputs.

The output $\mathcal{S}^*$ resides purely in the symbolic language space and is therefore independent of low-level visual observations. Formally, we define a two-stage factorization of the overall policy:

$$\pi(a_t \mid o_t, G) = \pi_{\text{exec}}(a_t \mid o_t, \mathcal{S}^*), \quad \mathcal{S}^* = \pi_{\text{plan}}(G),$$

where $\pi_{\text{plan}}$ maps high-level goals $G$ into symbolic plans, and $\pi_{\text{exec}}$ grounds these symbolic subtasks into action primitives $a_t$ conditioned on visual observations $o_t$, which will be described in detail in the following section w.r.t the multimodal planner.

This factorization highlights the modularity of our framework: the task-level planner is responsible for semantic reasoning in symbolic space, while mid-level multimodal planner and low-level controller modules handle grounding and execution. Such a separation improves robustness and reusability, especially in complex, sensor-rich scientific environments. We provide a detailed discussion on the feasibility of the decoupling manner in Appendix C.

## 3.3 MID-LEVEL MULTIMODAL PLANNER

The mid-level multimodal planner bridges high-level task decomposition and low-level control by enriching subtask execution with perceptual grounding. It consists of two complementary components: (1) a multimodal predictor that generates future scene representations conditioned on the current observation and task description, and (2) a multimodal language model (MLM) that translates predicted goals into structured subtask instructions. This design explicitly integrates visual and semantic cues, ensuring that subtask generation remains consistent and grounded in perceptual information.

### 3.3.1 VISUAL PREDICTION

The multimodal predictor is built on a conditional diffusion model (Du et al., 2023), trained to generate future motion frames for manipulation planning. To better capture the visual context between the initial and target states, the model predicts a full sequence of intermediate frames.

Task semantics are encoded by CLIP to obtain high-level representations of task instructions. UNet extracts hierarchical visual features from the input image, and cross-attention is employed to integrate CLIP embeddings with visual features, thereby aligning task semantics with image representations. The diffusion process gradually perturbs input data with noise, enabling robust

generation through denoising inference. The process of adding noise is defined as $q(i_t|i_{t-1}) = \mathcal{N}(i_t; \sqrt{1-\beta_t}i_{t-1}, \beta_t\mathbf{I})$, where $\beta_t$ is the predefined noise schedule, $1-\beta_t$ decides the proportion of original image information preserved at each diffusion step, $t = 1, 2, ..., T$ represents the timestep in the diffusion process. Correspondingly, $i_{t-1} = \frac{1}{\sqrt{1-\beta_t}}\left(i_t - \frac{\beta_t}{\sqrt{1-\bar{\alpha}_t}}\epsilon_\theta(i_t, t)\right) + \sigma_t z, z \sim \mathcal{N}(0, I)$ represents the reverse process that learns to denoise and gradually recover the original data distribution, where $\bar{\alpha}_t$ is the cumulative attenuation coefficient in the forward diffusion process, $\epsilon_\theta$ is the neural network–predicted noise, $\sigma_t$ is the noise standard deviation in the reverse process and $z$ is a standard normal random noise vector used to introduce stochasticity. Through the alignment and fusion of these multimodal features, the model can predict the future frame required for task execution. The generated target frame with the task description is used as the input of MLM. We assume that the multimodal predictor as $P_\theta$. The task description is $l$. The current environment frame as $i_0$, the frame at timestep $t$ is predicted as $i_t$, then the model should be like:

$$P_\theta(\text{UNet}(i_0, \text{CLIP}(l))) \rightarrow \{i_0, i_1, i_2, \ldots, i_t\}. \tag{1}$$

The loss function of the multimodal predictor adopts the same L2 loss as the diffusion model. However, to enhance model stability during training at various time steps $t$, we employ a loss reweighting strategy based on the Signal-to-Noise Ratio (SNR) (Ho et al., 2020), enabling dynamic adjustment of the loss across time. This strategy guides the model to concentrate on time steps which contain more informative signals. In the meantime, to address the issue of gradient explosion during training, we incorporate gradient clipping to ensure stable and reliable training.

$$L_{\text{final}} = \text{SNR}(t) \cdot \|\hat{\epsilon}_\theta(i_t, t) - \epsilon\|_2^2, \tag{2}$$

where $\epsilon$ means the real noise, $i_t$ means the noisy image at time step $t$.

To adapt to scientific experimental scenarios, we train the multimodal predictor using a dataset collected in a simulated laboratory environment. To reduce GPU memory consumption and enhance training efficiency, we crop out low-information regions from input images and set each training sequence to 8 frames per batch. The input resolution is fixed at $(256, 256)$, and the CLIP encoder is frozen during training to preserve pretrained semantics. A random frame from each sequence is selected as the initial observation. By predicting future frames, the multimodal predictor enhances the visual grounding of robotic actions, reduces collision risk, and improves interaction quality between the robotic arm and its environment. This provides rich visual-semantic cues that facilitate more accurate subtask grounding for downstream multimodal planning.

### 3.3.2 SUBASK DECOMPOSITION

Leveraging the strong visual reasoning and planning ability of MLMs, we further refine the subtasks into executable action primitives, thereby bridging high-level prompts to low-level control. Specifically, we use GPT-4o for subtask decomposition, where each subtask is mapped into a set of action primitives tailored to scientific environments: move, grasp, pour, and stir. To reduce ambiguity and computational overhead, we design a structured prompt format (Fig. 3) that conditions the MLM on paired visual inputs (current and predicted goal images) together with high-level task descriptions.

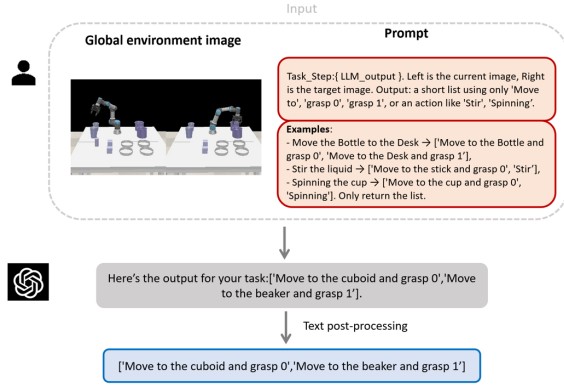

Figure 3: The example prompt for the MLM.

In this scheme, most subtasks are realized through move and grasp, with continuity enforced by concatenation (e.g., "move and grasp"), while pour and stir are treated as atomic operations directly invoked

Table 1: The scientific experiment dataset.

| Task | Description | Prompt template | #Training data |
|---|---|---|---|
| Pick_place | Basic object relocation. | 1. Move <object> to <object>.
2. Put <object> on <object>.
3. Place <object> next to <object>.
... | 88 |
| Pour | Single-step pouring of liquid. | 1. Pour <liquid>.
2. Pour <liquid> into <container>.
3. Pour <liquid> from <container>.
... | 25 |
| Stir | Stir the liquid in a container. | 1. Stir <liquid>.
2. Stir <liquid> in <container>.
3. Perform stirring.
... | 27 |
| Mix | Multi-step reactant mixing experiment. | 1. Conduct reactant mixing experiment.
2. Perform multi-step mixing.
3. Combine <liquid> and <liquid>.
... | 50 |
| Crystallize | Crystallization cultivation experiment. | Conduct crystallization.
Grow <object> from <solution>.
Perform crystallization cultivation.
... | 73 |

when needed. Each `move` is expressed as `move to <target>`, and each `grasp` as `grasp 1/0`, denoting grasp or release.

The MLM thus outputs a structured sequence of primitives, providing an interpretable and robot-compatible intermediate representation. This enables robust decomposition of complex long-horizon tasks while remaining lightweight enough for downstream reinforcement learning, where the robot interacts with the simulation environment to refine execution policies and acquire fine-grained skills with minimal collisions.

### 3.4 LOW-LEVEL CONTROLLER

To execute the action primitives, we use a continuous-control policy that maps observations to joint-level commands. While DDPG (Lillicrap et al., 2016) is one option, other continuous controllers could also be used. During training, exploration can be encouraged through noise injection.

The reward function balances task progress, success, and safety:

$$r = r_{\text{move}} + r_{\text{grasp}} - r_{\text{collision}}, \tag{3}$$

where $r_{\text{move}}$ encourages approaching the target, $r_{\text{grasp}}$ rewards successful grasps, and $r_{\text{collision}}$ penalizes collisions.

To support long-horizon tasks, the environment is reset only at task initialization or completion, maintaining continuity across subtasks. This allows the low-level policy to produce consistent behaviors over extended sequences.

### 3.5 SIMULATION SETUP

The simulation environment is built using the CoppeliaSim (VREP) simulation software and is designed to better suit the requirements of material scenarios. In the simulation environment, the UR3 robotic arm is used as the task execution mechanism. The environment includes experimental apparatus such as beakers and petri dishes, which are specifically tailored to material scenarios. Graspable objects in the environment, including cylinders, cubes, cups and sticks, represent different categories of experimental items. Notably, although our framework uses visual prediction, it focuses on understanding spatial relationships and task-relevant interactions, rather than recognizing detailed object appearances.

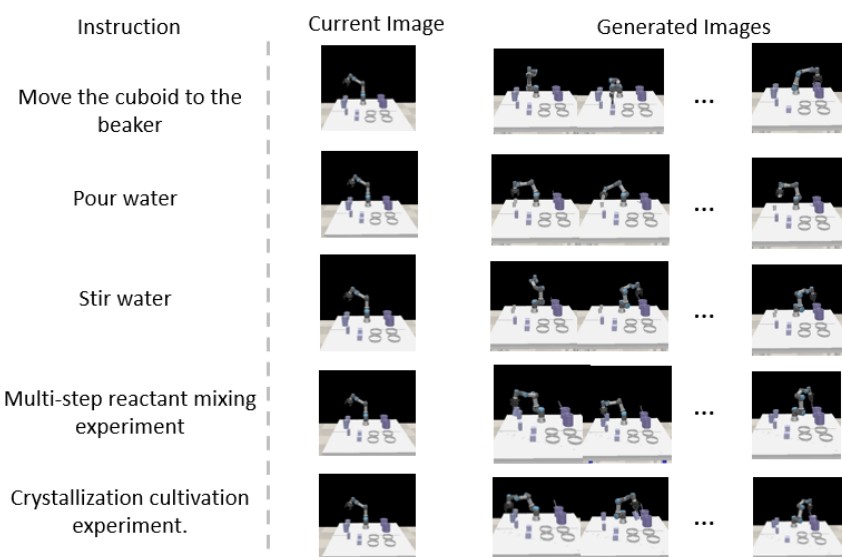

Figure 4: The prediction output of the multimodal predictor for action primitives.

The entire simulation environment is equipped with collision detection and distance sensing, allowing the DDPG model to directly access the current object states through the underlying functions of the environment, thereby updating the model's reward values. Also, we collected a representative dataset suitable for scientific experiments by using this simulation environment, which primarily consists of three types of long-horizon tasks and two types of extra-long-horizon tasks. Instead of using a single fixed instruction for each task, we design a prompt template pool for every task category. These template pools are composed of semantically equivalent but lexically diverse sentences that are randomly generated by a LLM, ensuring rich linguistic variability. The dataset demonstrates the execution steps of the robot to complete the target task. The size of each training data is shown in Table 1 The scale of the dataset is sufficient to train the multimodal predictor, enabling the acquisition of accurate prediction results. To ensure that our benchmark reflects the diversity and variability of real scientific workflows, we construct a large-scale test set in our simulation experiments. For each task, we randomly pair multiple GPT-generated instructions from the prompt-template pool with different object combinations to create 20 distinct input prompts. For each prompt, we further sample 50 randomized initial scenes with varying object poses and spatial layouts, resulting in 1,000 evaluation trials per task.

## 4 EXPERIMENTS

### 4.1 EXPERIMENTAL SETUP

We use single NVIDIA GeForce RTX 4090 GPU for model training and inference. In the multimodal predictor, we use the AVDC Flow Diffusion Model. To accommodate GPU memory constraints, the batch size is set to 1. Each batch contains 8 robot motion images corresponding to a single task category. Additionally, the image resolution is set to (256, 256). We utilize GPT-4o to perform environment understanding for the robotic system. For the reinforcement learning with DDPG, the replay buffer size is set to 4.

Table 2: Clarity and relevance of generated images.

| Task | Laplacian variance | CLIP Similarity |
|------|-------------------|-----------------|
| Pick_place | 1541.93 | 26.88 |
| Pour | 1625.44 | 23.18 |
| Stir | 1578.26 | 22.73 |
| Mix | 1613.07 | 26.96 |
| Crystallize | 1578.56 | 26.86 |

## 4.2 VISUALIZATION

To evaluate the quality of generated frames from the multimodal predictor, we assess both visual clarity and semantic alignment using standard quantitative metrics:

**Laplacian Variance** is used to measure image sharpness. It calculates the variance of the Laplacian of each frame, where higher values indicate clearer edge details.

**CLIP Similarity** quantifies the semantic alignment between textual instructions and predicted images. We extract embeddings from the pretrained CLIP model and compute cosine similarity between the instruction and the corresponding image frames.

Results are shown in Table 2. The generated frames consistently achieve Laplacian variance values above 1500, indicating sufficient visual sharpness to capture robot motion details. Additionally, the CLIP similarity scores remain above 21, reflecting strong semantic alignment between task descriptions and predicted visual content. This alignment helps guide the model toward generating more accurate and contextually grounded robot actions. We also show examples of predicted future frames of different tasks generated by the multimodal predictor in Fig. 4.

## 4.3 EXPERIMENTAL RESULTS ON THE SCIENTIFIC EXPERIMENT DATASET

We test the success rates of CoMP and other methods using the benchmark we designed.

In approaches that only rely on RL model, the DDPG model exhibits near-zero success rates on complex, long-horizon tasks due to the absence of task decomposition and planning mechanisms. Its performance is nearly indistinguishable from random trial-and-error, making meaningful comparison infeasible. The success rate of different methods results are shown in Table 3.

**LLM+RL (LLaMA-3.1-8B, w/o CoT).** In this variant, the multimodal predictor (MP) is removed and the mid-level MLM is replaced with LLaMA-3.1-8B. COT prompting is not used, so the model generates subtasks directly from the task prompt without stepwise reasoning to test the limitations of methods that only rely on language guidance. We incorporated environment descriptions into the prompt to guide the large language model in decomposing the input task based on the current environmental context. The resulting subtasks are then passed to the reinforcement learning model to evaluate task success rates.

**MLM+RL (LLaMA3.2, w/o CoT).** We use the MLM to demonstrate the importance of visual information input. Furthermore, we replace the MLM in CoMP to validate the rationality of our designed prompt in performing task decomposition and planning.

**LLM+RL (GPT-4o).** We use GPT-4o with the same COT prompting as our method for task decomposition. However, it does not receive any visual input. The model generates subtasks purely based on textual prompts describing the task and environment. This configuration tests how performance is affected when high-level reasoning is performed without visual grounding.

**MLM+RL (GPT-4o).** We remove the MP and the LLM with COT prompting maintained from CoMP. Compared with the MLM+RL (LLaMA3.2) baseline, the results demonstrate the reliability of our structured prompt design in MLM. Moreover, the noticeable drop in success rate when both the MP and task-level planner are absent further confirms the importance of these two parts.

**MP+MLM+RL (GPT-4o).** In CoMP, we only remove the task-level planner to evaluate its contribution. As shown in the table, the performance difference is relatively small for shorter tasks like `pick_place`, `pour` and `stir`. However, in the `mix` and `crystallize` tasks, task-level planner plays a significant role in improving success rates. This further demonstrates the robustness of the task-level planner in handling long-horizon tasks.

**CoMP.** We evaluated the CoMP model on a benchmark tailored for scientific experimental environments. The model demonstrated strong performance in the `pick_place` task. However, in tasks such as `pour` and `stir`, the success rate declined slightly due to the difficulty in detecting transparent cup walls and thin stir bars, making them more challenging than simple object grasping. For long-horizon tasks like the multi-step reactant mixing experiment and the crystallization cultivation experiment, the success rate also decreased compared to short-horizon tasks, owing to

Table 3: Success rate (%) of different methods on the scientific experiment dataset.

| Task | LLM+RL (Llama-3.1-8B w/o COT) | MLM+RL (LLaMA3.2 w/o CoT) | LLM+RL (GPT-4o) | MLM+RL (GPT-4o) | MP+MLM+RL (GPT-4o) | CoMP |
|---|---|---|---|---|---|---|
| Pick_place | 39.6 | 45.8 | 70.4 | 78.9 | 79.3 | 80.3 |
| Pour | 34.8 | 44.7 | 54.6 | 62.6 | 62.4 | 64.5 |
| Stir | 30.2 | 39.6 | 34.8 | 70.5 | 70.8 | 74.4 |
| Mix | 23.3 | 35.5 | 23.8 | 43.2 | 43.6 | 52.3 |
| Crystallize | 11.6 | 12.9 | 12.4 | 21.0 | 21.5 | 30.1 |

Table 4: Success rate (%) of different methods on RLBench.

| Task | PerAct | OpenVLA | CoT-VLA[*] | MoLe-CogAct | CoMP |
|---|---|---|---|---|---|
| Pick cup | 40.3 | 85.6 | 86.1 | 87.0 | 48.6 |
| Push button | 48.6 | 84.6 | 85.0 | 88.4 | 82.8 |
| Take umbrella | 20.2 | 28.5 | 27.9 | 36.3 | 42.2 |
| Put knife | 16.7 | 8.4 | 8.7 | 24.7 | 72.3 |
| Put money | 44.7 | 25.3 | 25.7 | 31.9 | 46.5 |

[*]Re-implemented following the paper, as official code is unavailable.

the extended visual and procedural complexity. Nonetheless, CoMP consistently achieved the best overall performance, validating the necessity and effectiveness of each module in the system.

From the experimental results, we observe the critical role of visual information and CoT reasoning structure within the CoMP framework. Moreover, each module in CoMP proves to be indispensable. The success rate of CoMP in scientific experiment tasks meets the practical requirements of such experiments.

### 4.4 EXPERIMENTAL RESULTS ON THE PUBLIC DATASET

We also evaluate our method on a public dataset, RLBench (James et al., 2020), which is an open-source robotic manipulation benchmark suite that provides over 100 diverse and realistic tasks for robot learning.

To differentiate CoMP from CoT-VLA, and owing that the code of CoT-VLA is not publicly available, we reproduced the entire pipeline based on the VILA-U model (Wu et al., 2025) described in the CoT-VLA paper. Considering that CoT-VLA first generates high-level subtasks and then low-level trajectories, which CoMP also follows a hierarchical structure. If sufficient training data are available, CoMP's mid-level multimodal planner and low-level controller can be replaced with CoT-VLA, while retaining the task-level planner. For experiments using the VILA-U model, we fine-tuned it directly on the same collected training data as OpenVLA, without pretraining on robot demonstrations or action-less videos. This decision is made because pretraining on real-world data could potentially degrade the model's performance in the simulated environment.

We select 5 long-horizon tasks that closely align with the typical procedures of scientific experiments in RLBench to validate the performance of CoMP, and compare the success rate with PerAct (Shridhar et al., 2022) , OpenVLA (Kim et al., 2024), COT-VLA (Zhao et al., 2025) and MoLe-CogAct (Zhang et al., 2025) methods. As shown in Table 4, CoMP achieves a higher success rate on long-horizon tasks, demonstrating better adaptability to various scenarios of long-horizon tasks by benefiting from more world knowledge with an additional LLM.

### 4.5 EXPERIMENTAL RESULTS ON SCIENTIFIC EXPERIMENT IN REAL-WORLD SCENARIO

We designed a sim-to-real experimental setup where the real objects differ in shape from those in simulation but share the same sizes and positions. To obtain reliable object poses, we fine-tuned a YOLOv8 detector on real laboratory instruments. We combined it with a D415 depth camera to acquire 3D coordinates that can be directly mapped into simulation. This lightweight calibration

Table 5: Extended real-world scientific experiment tasks with human demonstrations.

| Task | Description | Prompt template | #Training data |
|------|-------------|-----------------|----------------|
| Weigh | Place containers on the balance for automated weighing. | 1. Weigh <object>. 2. Weigh <object> with the balance. ⋯ | 27 |
| Shake | Load the container onto the shaker. | 1. Load <container> onto the shaker. 2. Shack <container>. ⋯ | 25 |

allows CoMP to be deployed without any model fine-tuning on the robot side. CoMP achieved success rates of 76.3%, 62.1%, 70.8%, 50.6% and 29.4% on the Pick_place, Pouring, Stir, Mix and Crystallize tasks, respectively, in real-world scenarios.

To further demonstrate CoMP's flexibility in scenarios where certain laboratory tools are difficult to model in simulation, we additionally introduce two new tasks that rely on real human-executed experiment videos as demonstrations. Task definitions and training statistics are summarized in Table 5. Even in these challenging settings, CoMP maintains strong performance, achieving 66.7% on Weigh and 70.0% on Shake.

Further details and visualizations are provided in Appendix I, including simulated and real-world results, human demonstrations and robot executions, as well as analysis of failure cases.

### 4.6 MODULE EFFICIENCY AND RUNTIME

Most components of our method do not require training. All LLM/MLM modules are used off-the-shelf, while only the diffusion-based MP and RL controller need training. Among these, the diffusion model is the most expensive, and the RL policy can even be trained on a single PC GPU (e.g., RTX 4060). Inference latency for each module is shown in Table 6. Except for RL, all reasoning can be completed before execution. The predictor and MLM can optionally update during execution to reduce error accumulation, though this is not enabled here. Given that subtasks last seconds, MP resolution can be reduced to $128{\times}128$ for efficiency.

| Component | Latency (ms) | Runs Online? |
|-----------|--------------|--------------|
| LLM | 3024 | No |
| MP | 5714 | Optional |
| MLM | 3441 | Optional |
| RL | 15 | Yes |
| **Online total** | 15 | - |

Table 6: Per-module inference latency and online/offline execution.

## 5 CONCLUSION

We presented CoMP, a compositional and decoupled framework for long-horizon robotic planning in scientific experiments. By combining task-level CoT decomposition, multimodal visual prediction, and RL-based control, CoMP enables transparent, modular reasoning without the need for task-specific fine-tuning. Experiments on both our proposed benchmark and RLBench demonstrate consistent improvements in success rate, procedural correctness, and generalization over strong baselines.

While our current setup employs simplified geometric proxies for laboratory objects, limiting realistic grasping and manipulation, future work will incorporate grasp pose estimation and diverse lab materials to improve real-world applicability.

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

APPENDIX

## A  LLM USAGE STATEMENT

Large Language Models (LLMs) are used to assist in language polishing and improving the readability of the manuscript.

## B  DETAILED DESCRIPTION OF THE SELF-VERIFICATION OPERATIONS

**Subgoal Validation.**  The LLM is prompted to check whether each subtask aligns with the overall task objective through natural-language self-reflection and identify potential contradictions, redundancies, or omissions.

**Constraint Consistency Check.**  For each subtask, we guide the model to ensure each subtask satisfies key physical, temporal, and causal constraints, detecting invalid ordering (e.g., measuring before synthesis) or mismatches (e.g., heating before reagent addition).

**Correction and Refinement.**  When inconsistencies are identified, the model is prompted to revise and revalidate subtasks until the sequence is logically coherent and operationally feasible.

## C  DISCUSSION OF WHY AND WHEN DECOUPLING IS VALID

In this section, we provide a brief theoretical analysis of the conditions under which the decoupling between the task-level symbolic planner and downstream execution is valid. We focus on the subtask sequence generated by the task-level planner and analyze its contribution to modularity, while noting that downstream action primitives and RL policies do not affect this argument.

**Preliminaries.**  Let $\mathcal{G}$ denote a high-level task goal, $\mathcal{S} = (s_1, s_2, \ldots, s_T)$ the symbolic subtask sequence generated by the task-level planner, and $\pi$ a low-level execution policy acting on continuous observations $o_{1:T}$. Task success is represented by the event $\mathcal{E} = \{\text{goal achieved}\}$.

**Lemma 1** (Task-symbolic sufficiency). *If the execution policy $\pi$ depends only on observations $o_{1:T}$ given $\mathcal{S}$, i.e.,*

$$P(\pi \mid \mathcal{S}, \mathcal{G}, o_{1:T}) = P(\pi \mid \mathcal{S}, o_{1:T}),$$

*then the goal $\mathcal{G}$ influences execution only through the symbolic plan $\mathcal{S}$.*

**Lemma 2** (Conditional independence). *If symbolic decompositions and observations are conditionally independent given the goal,*

$$P(\mathcal{S}, o_{1:T} \mid \mathcal{G}) = P(\mathcal{S} \mid \mathcal{G}) \, P(o_{1:T} \mid \mathcal{G}),$$

*then $\mathcal{S}$ carries no redundant information about $o_{1:T}$ beyond $\mathcal{G}$.*

**Proposition 1** (Factorization of success probability). *Under Lemmas 1 and 2, the probability of success can be factorized as*

$$\mathcal{E} := \max P(\pi \mid \mathcal{G}, o_{1:T}), \tag{4}$$

$$P(\pi \mid \mathcal{G}, o_{1:T}) = \sum_{\mathcal{S}} P(\pi, \mathcal{S} \mid \mathcal{G}, o_{1:T}) \tag{5}$$

$$= \sum_{\mathcal{S}} P(\pi \mid \mathcal{S}, o_{1:T}) \, P(\mathcal{S} \mid \mathcal{G}). \tag{6}$$

*Hence the contribution of reasoning (via $P(\mathcal{S} \mid \mathcal{G})$) is* decoupled *from perception and control (via $P(\mathcal{E} \mid \mathcal{S}, o_{1:T})$).*

This shows that improvements in task-level reasoning and low-level execution independently contribute to overall success, consistent with modular planning frameworks (Kaelbling et al., 1998).

**When decoupling is valid.** Decoupling holds if:

- Each subtask $s_i$ contains sufficient symbolic information for execution without direct dependence on $\mathcal{G}$;
- Observations and subtasks are approximately conditionally independent given $\mathcal{G}$;
- Subtasks capture the causal structure of the task relevant for achieving the goal.

For example, for a task like "mix reagents A and B", the task-level planner may output subtasks such as *add reagent A* and *add reagent B*. Each subtask carries enough symbolic information for downstream execution, the low-level observations for executing these actions are largely independent of the high-level goal, and the sequence preserves the necessary causal order. Compared to a fully end-to-end approach that maps from goal and observations directly to actions, decoupling reduces the complexity and naturally aligns with the modular structure of the task.

Overall, while the conditions are not guaranteed in all settings, they are relatively easy to satisfy in well-structured tasks, which explains why decoupling is a practical and effective design choice.

**When decoupling may fail.** In practice, these assumptions may be violated in cases such as:

- If $\mathcal{S}$ omits crucial physical requirements (e.g., exact forces or contact stability), then $P(\mathcal{E} \mid \mathcal{S}, o_{1:T})$ still implicitly depends on $\mathcal{G}$.
- If $\mathcal{S}$ is entangled with perception (e.g., symbolic labels directly derived from raw visual embeddings), then independence may fail.
- If success depends on hidden histories not captured in $\mathcal{S}$, factorization does not hold.

**Fallback as constrained refinement.** Execution failures can be modeled as the discovery of additional constraints $C$ on symbolic decompositions. Formally,

$$\mathcal{S}^{(k+1)} \sim P(\mathcal{S}^{(k)} \mid \mathcal{G}, C),$$

where $C$ is induced from observed failure conditions (e.g., contact instability). If the constraint set $\mathcal{C}$ is finite and each refinement step reduces the feasible set of $\mathcal{S}$, then iterative fallback converges in at most $|\mathcal{C}|$ iterations: either a feasible $\mathcal{S}^\star$ is found, or infeasibility is proven. This parallels refinement strategies in symbolic AI.

**Limitations and outlook.** Our current system does not implement fallback at this stage, i.e., the planner runs only once. Nonetheless, the above analysis suggests a principled extension: failed executions could be recycled as symbolic constraints to refine subtask decomposition. Future work may explore this integration, which would tighten the link between symbolic reasoning and embodied feedback while preserving modularity.

## D    TRAINING DETAILS

We implemented our multimodal predictor using PyTorch. The backbone of the diffusion model is U-Net. The multimodal predictor is conducted with the following configuration:

- Number of diffusion steps: 100
- Sampling steps: 100
- loss: L2
- SNR-based loss weighting: True
- Image size: $(256, 256)$

We adopt OpenAI's CLIP as the text encoder, with its parameters kept frozen throughout training to preserve the pretrained semantic representations. The multimodal predictor is trained on an NVIDIA RTX 4090 GPU, and the model checkpoints are saved every 2,500 iterations. The training is conducted with the following configuration:

- Learning rate: 1e-5
- Batch size: 1
- Image sequence length: 8
- Total training steps: 60,000
- Precision: Mixed precision (fp16)

The training process of multimodal predictor takes approximately three days on our collected dataset.

## E    RESULTS OF DIFFERENT LARGE MODELS

Since only relying on large language models to evaluate outputs can be inaccurate, which often leading to redundant content. Thus we assessed the performance of different LLMs and MLMs on our defined tasks through human evaluation. For each model, we prompt the large model to generate 20 task decompositions in each round, and compute the corresponding success rate based on expert evaluations. The final success rate reported in the table is the average of 5 rounds. We use LLaMA3.1 as the LLM and GPT-4o as the MLM in our framework.

Table 7: Success rate (%) of different methods on RLBench

| Model | Pick_place | Pour | Stir | Mix | Crystallize |
|---|---|---|---|---|---|
| Qwen3 | 46.3 | 34.8 | 100 | 87.4 | 55.6 |
| Deepseek-V3 | 99.5 | 46.6 | 100 | 32.8 | 90.5 |
| Llama3.1 (ours) | 100 | 74.6 | 100 | 72.4 | 77.5 |

Table 8: Success rate (%) of different MLMs

| Model | Pick_place | Pour | Stir | Mix | Crystallize |
|---|---|---|---|---|---|
| Qwen2.5 | 76.7 | 75.5 | 76.4 | 73.4 | 72.5 |
| Llama3.2 | 82.3 | 81.2 | 81.7 | 78.7 | 77.6 |
| GPT-4o (ours) | 100 | 100 | 100 | 99.5 | 99.6 |

Table 9: Success rate (%) of CoT reasoning and its ablations (w/o Step1–4)

| Task | CoT | CoT w/o Step1 | CoT w/o Step2 | CoT w/o Step3 | CoT w/o Step4 |
|---|---|---|---|---|---|
| Pick_place | 100.0 | 44.5 | 37.8 | 46.7 | 41.3 |
| Pour | 74.6 | 13.7 | 23.7 | 24.8 | 22.5 |
| Stir | 100.0 | 57.2 | 28.6 | 78.5 | 61.4 |
| Mix | 72.4 | 42.0 | 24.5 | 65.7 | 12.8 |
| Crystallize | 77.5 | 71.7 | 51.3 | 54.8 | 30.2 |

## F    TASK DETAILS

Our collect dataset contains 5 types of tasks, which include `Pick_place`, `Pour`, `Stir`, `Mix` and `Crystallize`. To support these tasks, we also design 4 action primitives: `Move`, `Grasp`, `Slant` and `Stir`, each embodying representative features of scientific experimental procedures. The size of the collected data is $(512, 512)$.

- `Pick_place`: The `pick_place` task includes 3 types of object manipulation tasks: moving the cuboid to the beaker, moving the cuboid to the glass garden, and moving the cylinder

to the beaker. In these tasks, we use cuboids and cylinders to represent objects with different shapes and graspable properties. Each task contains approximately 30 images, which serve as motion guidance for the robot during execution. Fig. 10 illustrates example images from the `pick_place` task.

- `Pour`: Considering the specific procedural requirements of certain scientific experiments, we collected a visual demonstration set for the `pour` task. This task includes 25 images capturing the robot performing the pouring operation. We use a `cup` as the manipulable object to represent the action of pouring. The execution of the `pour` operation requires prior completion of the `move` and `grasp` operations from the `pick_place` task.

- `Stir`: The `stir` task is designed to perform liquid stirring operations. A thin rectangular block is used as a graspable stirrer to ensure successful execution of the `stir` operation. Similar to `pour`, the `stir` operation also requires prior completion of the `move` and `grasp` operations.

- `Mix`: Mix is a longer-horizon task. Considering the frequent use of reactant mixing in scientific experiments, we incorporate this task into our setup. The total number of training images for this task is 58. This long-horizon task can be decomposed and accomplished through a combination of the three previously defined action primitives.

- `Crystallize`: The `crystallize` task has a similar horizon length to the `mix` task, both being more complex than the former 3 tasks. It contains a total of 60 training images. This task can also be decomposed into a sequence of our designed action primitives. Unlike the `mix` task, however, the `crystallize` task requires more frequent use of `pick_place` operations rather than `pour`.

Table 10: Error rates (%) of E1 and E2 under CoT reasoning and its ablations (w/o Step1–4)

| Task | CoT | | CoT w/o Step1 | | CoT w/o Step2 | | CoT w/o Step3 | | CoT w/o Step4 | |
|---|---|---|---|---|---|---|---|---|---|---|
| | E1 | E2 | E1 | E2 | E1 | E2 | E1 | E2 | E1 | E2 |
| Pick_place | 0.0 | 0.0 | 55.5 | 27.8 | 62.2 | 11.7 | 42.7 | 43.1 | 59.7 | 19.9 |
| Pour | 2.6 | 25.4 | 86.3 | 19.2 | 76.3 | 9.5 | 70.2 | 35.2 | 77.5 | 29.1 |
| Stir | 0.0 | 0.0 | 42.8 | 25.7 | 67.3 | 35.8 | 14.3 | 7.2 | 38.6 | 0.0 |
| Mix | 7.1 | 27.6 | 58.0 | 48.3 | 9.4 | 75.5 | 8.6 | 34.3 | 0.0 | 87.2 |
| Crystallize | 8.0 | 14.5 | 0.0 | 28.3 | 18.6 | 37.1 | 7.1 | 35.4 | 19.9 | 69.8 |

## G  CoT Prompt design

Our CoT reasoning of the LLM is designed in 4 steps. Each step guides the LLM to think through how to properly decompose the long-horizon task and evaluate whether the resulting short-horizon subtasks are sufficient to accomplish the original objective. This decomposition process does not require visual input.

Instead, the model relies on commonsense reasoning, following the structured CoT prompts to complete the task breakdown. The complete reasoning chain is illustrated in Fig. 6a.

We also conducted distillation experiments on each of the four stages to demonstrate that they are essential components in the task decomposition process. The success rate is evaluated in the same manner as the section of "Results of Different Large Models".

We evaluated the task decomposition success rate of CoT reasoning when each module is ablated. The result is shown in Table 9.

Meanwhile, we defined two common types of errors for analysis: Step Redundant Error (E1) and Step Logical Error (E2), allowing for a more detailed evaluation of decomposition quality. We further record the occurrences of two defined error types, which are shown in Table 10. We use the crystallization cultivation experiment as an example to illustrate the two error types: E1 and E2.

1. Pick up cuboid (pick_and_place) → Logical purpose: Prepare for placing in beaker
2. Place cuboid in beaker (pick_and_place) → Logical purpose: Ensure cuboid is in the beaker
3. Pick up cup (pick_and_place) → Logical purpose: Prepare for pouring
4. Pour contents from cup into beaker (pour) → Logical purpose: Transfer liquid to beaker
5. Pick up stir stick (pick_and_place) → Logical purpose: Prepare for stirring
6. Stir the liquid (stir) → Logical purpose: Finish the task

(a) Correct CoT output

"1. Pick up cuboid (pick_and_place) → Logical purpose: Prepare for placing near the beaker
2. Place cuboid near the beaker (pick_and_place) → Logical purpose: Position for future use
3. Pick up cup (pick_and_place) → Logical purpose: Prepare for pouring
4. Pour contents from cup into beaker (pour) → Logical purpose: Transfer experimental liquid
5. Pick up cuboid (pick_and_place) → Logical purpose: Prepare for grasping and moving
6. Pick up stir stick (pick_and_place) → Logical purpose: Prepare for stirring
7. Place cuboid on stir stick (pick_and_place) → Logical purpose: Secure the cuboid for stirring"

(b) E1 error example: redundant step

"1. Pick up cuboid (pick_and_place) → Logical purpose: Prepare for movement
2. Move cuboid to the beaker (slant) → Logical purpose: Position cuboid for subsequent steps
3. Pick up cuboid (pick_and_place) → Logical purpose: Prepare for pouring
4. **Pour contents from cuboid** into the beaker (pour) → Logical purpose: Prepare for stir
5. Pick up stir stick (pick_and_place) → Logical purpose: Prepare for stirring
6. Stir the liquid (stir) → Logical purpose: Finish the task"

(c) E2 error example: incorrect logic

Figure 5: Examples of CoT module outputs: (a) correct, (b) E1 error, (c) E2 error.

Table 11: Success rate (%) of CoMP in real-world scenario

| Task | Pick_place | Pour | Stir | Mix | Crystallize | Weigh | Shake |
|------|-----------|------|------|-----|-------------|-------|-------|
| CoMP | 76.3 | 62.1 | 70.8 | 50.6 | 29.4 | 66.7 | 70.0 |

The correct CoT module output for the crystallization cultivation experiment is shown in Fig. 5a. We also provide example output images illustrating E1 and E2 errors, which are shown in Figs. 5b and 5c. In the E1 example, the model repeatedly outputs the instruction "pick up cuboid". Although the overall sequence is structurally complete, the redundancy in a single step leads to its classification as an E1 error. In the E2 example, the model outputs "pour contents from cuboid" when there is no object inside the cuboid. This results in a logically incorrect step, thus leading to its classification as an E2 error.

## H  MLM PROMPT DESIGN

The prompt design for the multimodal large model is primarily aimed at mapping the subtask instructions generated by the large language model to our predefined three action primitives, in order to guide the robot's motion accordingly. The detailed prompt structure is illustrated in Fig. 6b.

## Task:\"[question]\".
Environment Rules (Important Constraints):
- The cup contains the liquid, which needs to be Poured in the beaker.
- The cuboid is the solid of the experiment, which only can Pick_and_place.
- The beaker is immovable and ungraspable.
- The Stir operation requires to grasp the stick.
## Chain-of-Thought Reasoning:
Step 1: Understand the task objective  Thinking: What is the final goal we need to achieve? What would constitute success?
Step 2: Identify logical prerequisites  Thinking: What must be true before we start? What objects must exist and what are the constraints we must obey?
 Step 3: Decompose into primitive operations  Thinking: What sequence of basic operations (pick_and_place, pour, stir, slant) would accomplish this task **while strictly following all environment rules**?
Step 4: Validate the operation sequence  Thinking: Is this sequence logically complete and compliant with the constraints? Would it successfully accomplish the task?
## Final Output Format:Only output the operation sequence using this format:1. [First operation] → Logical purpose: [Why this step is necessary]  2. [Second operation] → Logical purpose: [Why this step is necessary]  ... n. [Final operation] → Final outcome: [How this completes the task] Be sure not to include any actions that grasp or move the beaker. Use only legal, necessary operations under the constraints Example 1: Mixing solutions from two beakers Task: Mix solutions from two beakers on the table. Chain-of-thought analysis: Step 1: Understand the task objective Thinking: The goal is to combine the contents of two beakers to create a mixed solution. Success means having all liquid combined in one container. Step 2: Identify logical prerequisites Thinking: We must have two beakers containing solutions. They must be accessible for manipulation. Step 3: Decompose into primitive operations Thinking: Logically, we need to move one solution into the other, then ensure they're properly mixed. This requires picking up one beaker, pouring its contents into the other, and then stirring. Step 4: Validate the operation sequence Thinking: This sequence covers all necessary steps: transferring the liquid and ensuring proper mixing. The task would be successfully completed.Operation sequence: 1. Pick up cup (pick_and_place) → Logical purpose: Prepare for pouring 2. Pour contents in the beaker(pour) → Logical purpose: Prepare for stir 3. Pick up one cup (pick_and_place) → Logical purpose: Prepare for pouring 4. Pour contents in the beaker(pour) → Logical purpose: Prepare for stir 5.Stir the liquid (Stir) → Logical purpose: Finish the task. Only give the Operation sequence.

(a) Design of the CoT prompt

You are given:
1. The LLM's initial task decomposition
2. The target task
3. Left image = global environment, right image = target state. The LLM's output may contain logical or procedural errors.
Your job:
1. Correct the decomposition so it is logically valid.
2. Break each step into the only primitive actions: Move, Grasp, Slant, Stir.
3. Use the following special rules: - pick_and_place → ['Move to <Object> and grasp 0', ''Move to <Object> and grasp 1'], - Pour → ['Move to <Object> and grasp 0', 'Slant', 'Place on the desk'], - Stir → ['Move to Stick and grasp 0', 'Stir']
4. Output only a Python list of actions.
Environment Rules(Important Constraints):
- The beaker is immovable and ungraspable.
- The cup contain the liquid, which need to slant in the beaker.
- The cuboid represents the solid of the experimental, which only can move and grasp.
- The stirring operation requires grasping a stir stick.
Example 1: Task: Move the cuboid to the beaker. LLM: 1. Pick up cuboid (pick_and_place) → Logical purpose: Prepare for placing in beaker 2. Place cuboid in beaker (pick_and_place) → Logical purpose: Finish the task. Output should be: ['Move to the cuboid and grasp 0', 'Move to the beaker and grasp 1'].
Example 2: Task: Pour the liquid. LLM: 1. Pick up cup (pick_and_place) → Logical purpose: Prepare for pouring 2. Pour the liquid to the beaker → Logical purpose: Finish the task. Output should be: ['Move to the cup and grasp 0', 'Move to the beaker','Slant'].
The LLM's initial task decomposition: \"[llm]\". The target task: \"[task]\". Give the output. Remember the enviroment rules. The output should be merge into a single list. Reduce the length.

(b) Design of the MLM prompt

Figure 6: Prompt designs for different reasoning strategies: (a) CoT and (b) MLM.

# I  EXPERIMENT IN REAL-WORLD SCENARIO

We replicate the experimental setup from the simulation environment in a real-world setting and evaluate the success rate using our proposed method. The results show in Table 11. The success rate in the physical environment is slightly lower than that in simulation, primarily due to inaccuracies in YOLOv8 detection and depth sensing. Although these issues could be mitigated by employing higher-precision 3D sensors, addressing them falls outside the scope of this work. The modular design of CoMP offers strong transferability, enabling flexible adaptation and application across diverse scenarios. We present the experimental images of real-world scenario in tasks such as Pick_place, as shown in Fig. 7 - 20. Notably, our framework allows flexible task skipping based on real-time conditions. In the weigh task, when the beaker is already on the balance, CoMP can bypass this subtask without additional actions. We additionally analyze common sim-to-real failure cases in Figs. 17 and 20. In the first case, the robot collides with an unseen object because the detector fails to recognize it. This object has not been included in our YOLOv8 fine-tuning set, leading to a missing detection and an incorrect motion plan. In the second case, the robot fails to grasp the target object due to inaccurate depth-based localization from the D415 sensor, resulting in insufficient gripper contact during execution.

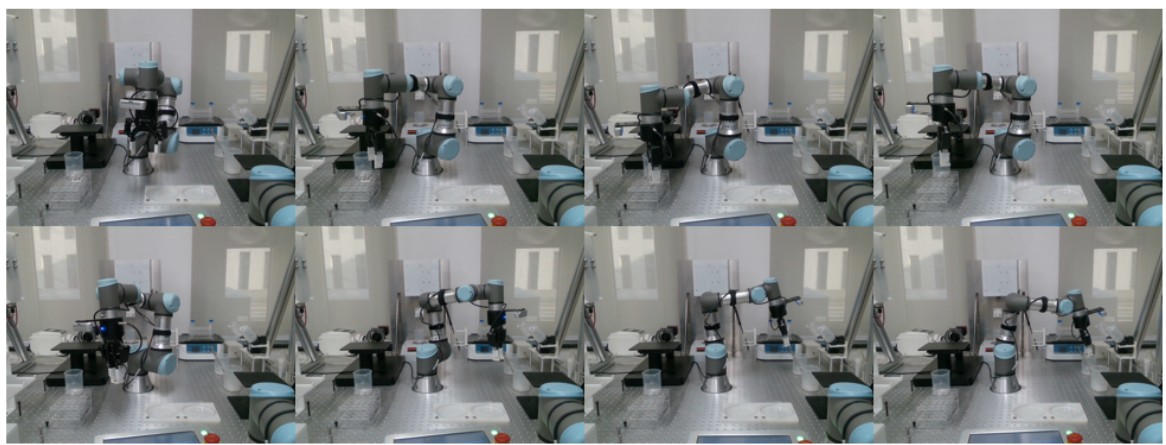

Figure 7: "Pick_place" task in real-world scenario.

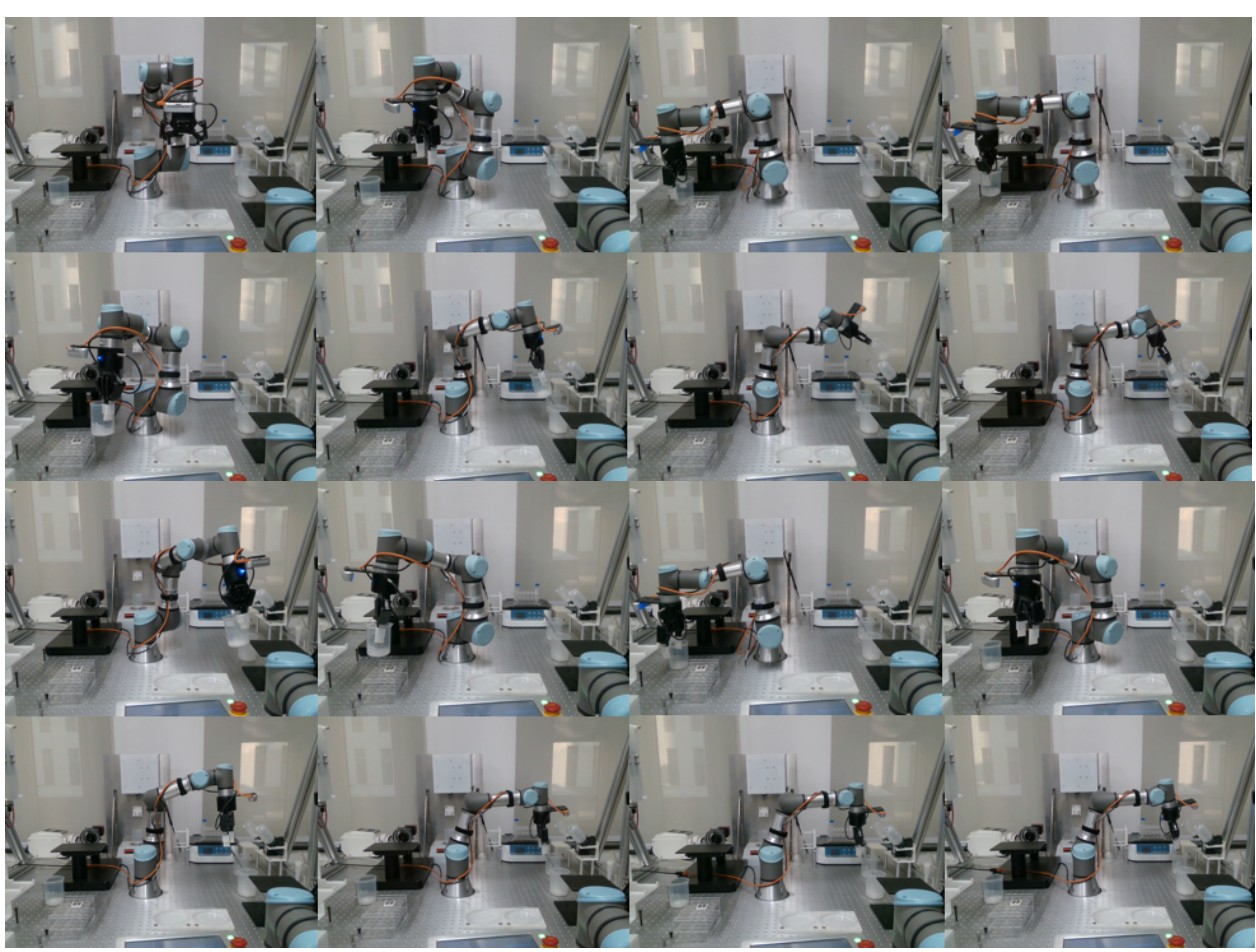

Figure 8: "Crystallize" task in real-world scenario.

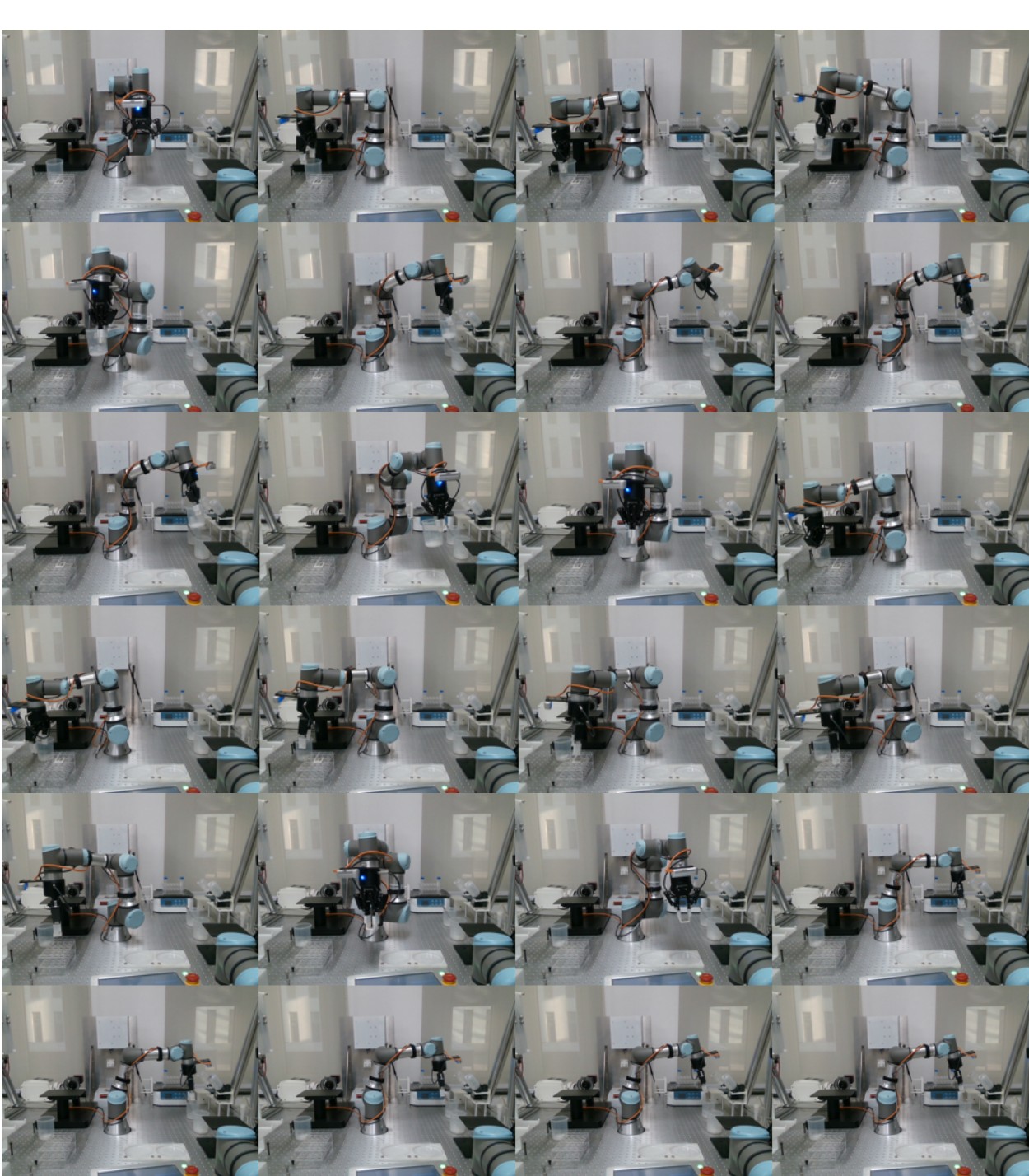

Figure 9: "Mix" task in real-world scenario.

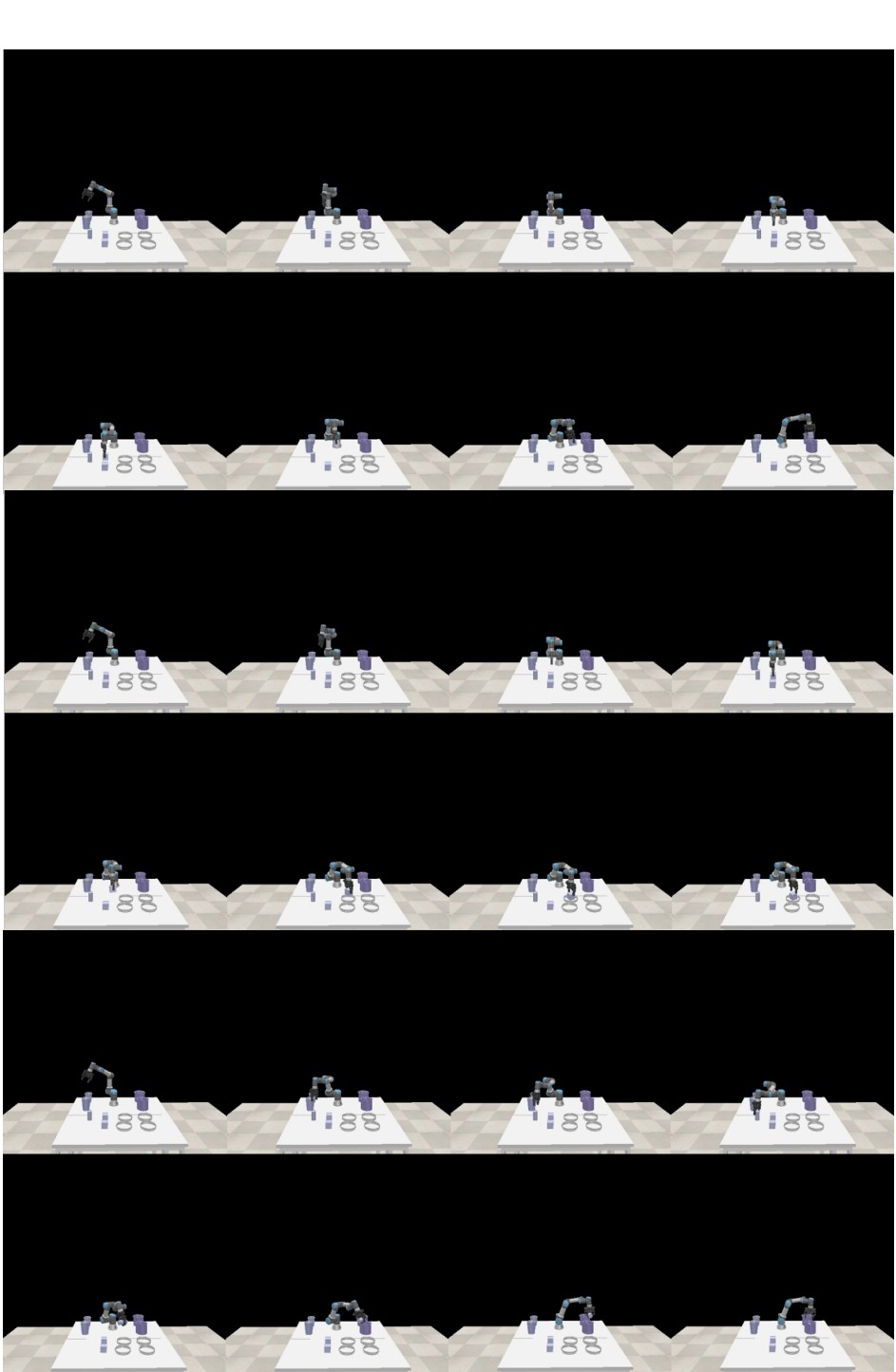

Figure 10: Visualization of the "pick_place" task.

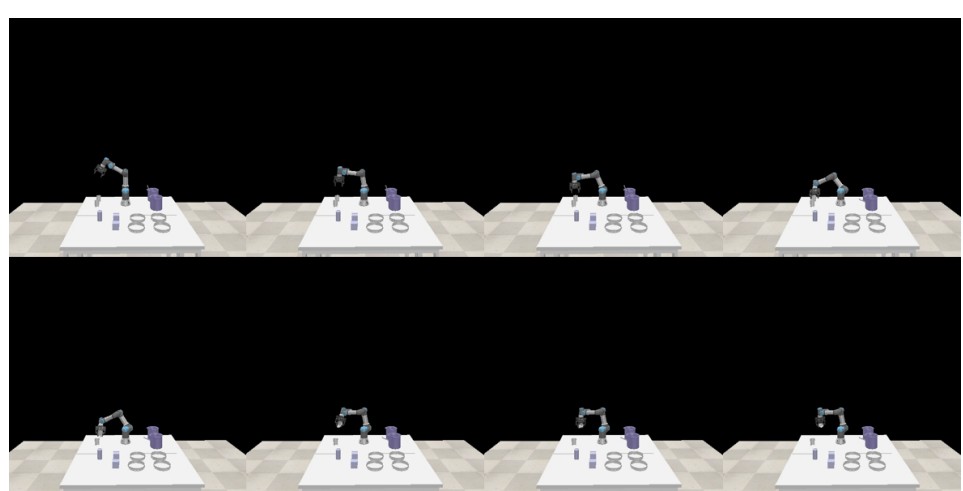

Figure 11: Visualization of the "pour" task.

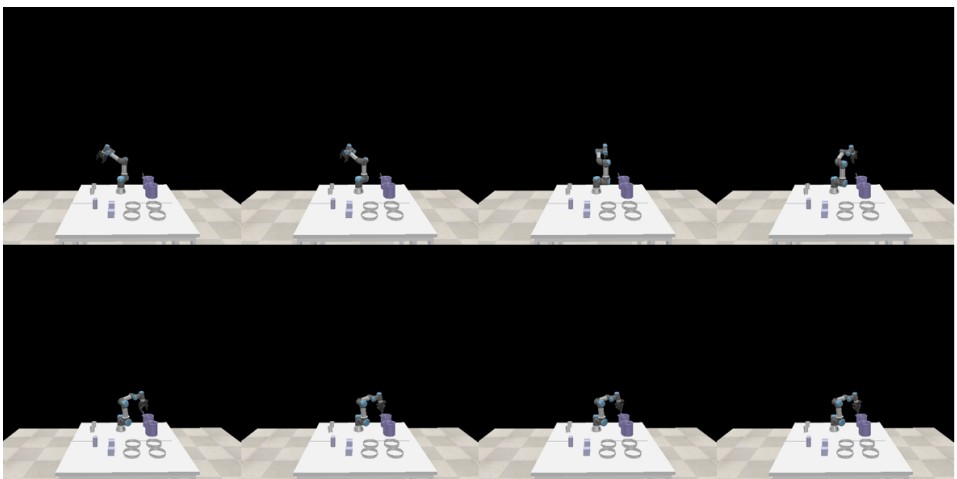

Figure 12: Visualization of the "stir" task.

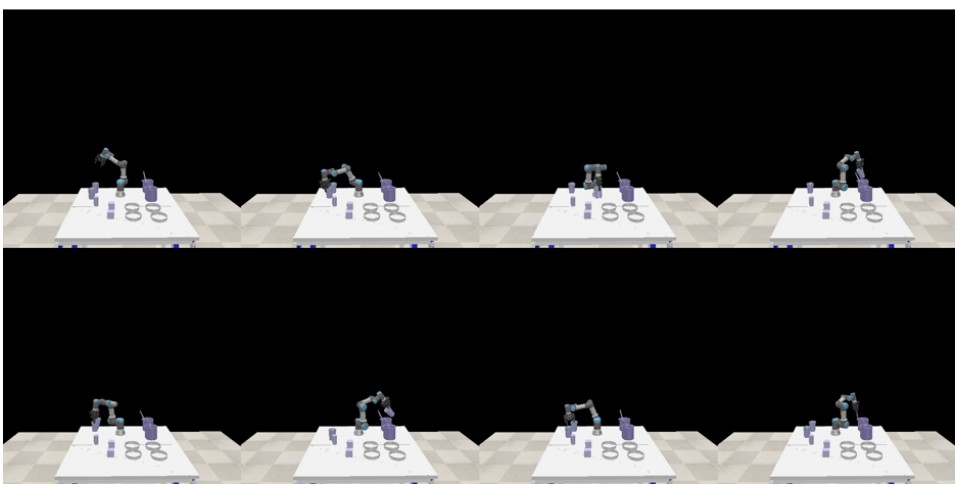

Figure 13: Visualization of the "mix" task.

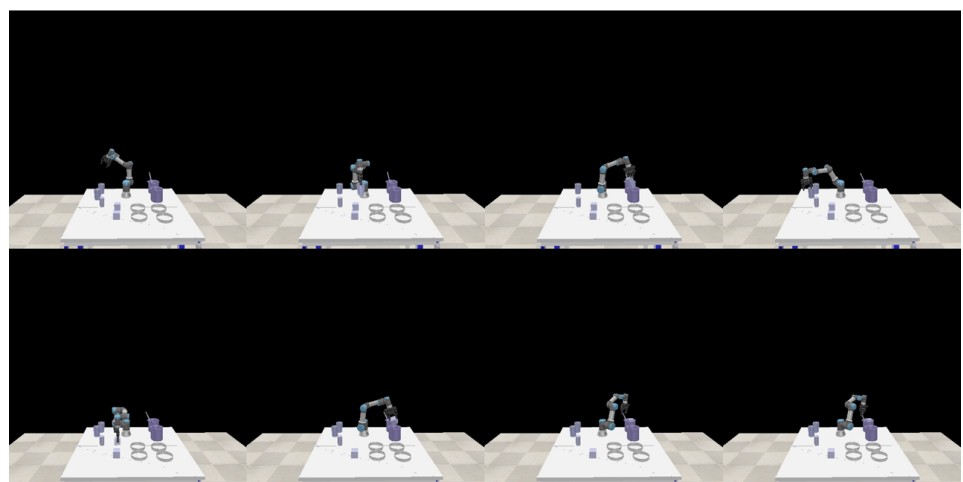

Figure 14: Visualization of the "crystalize" task.

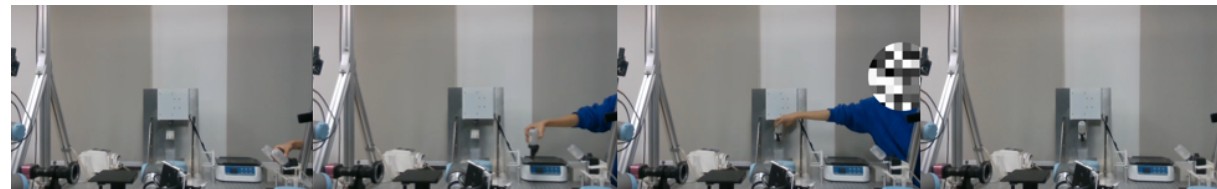

Figure 15: Human demonstration of the "weigh" task in real-world scenario.

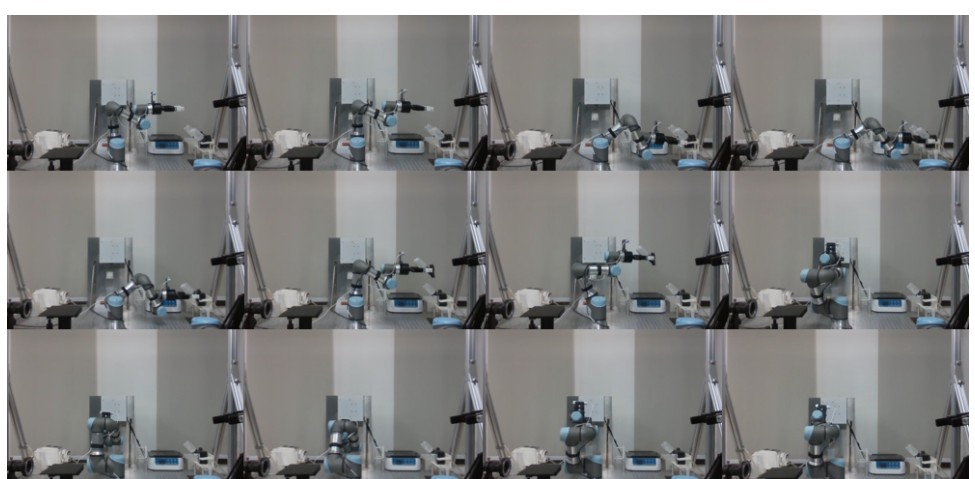

Figure 16: "Weigh" task in real-world scenario.

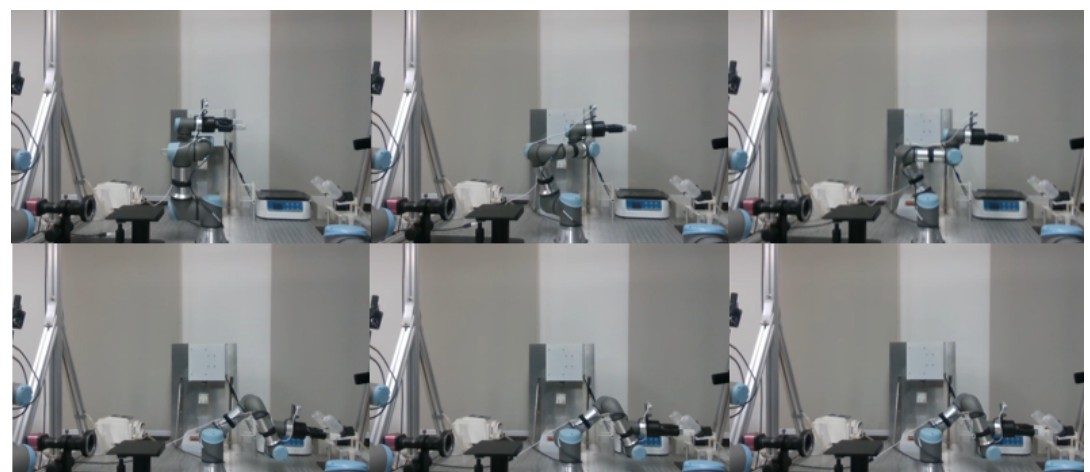

Figure 17: Failure case of the "weigh" task in real-world scenario.

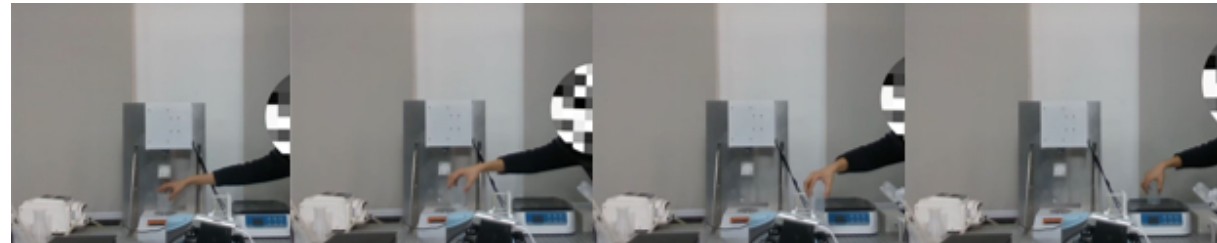

Figure 18: Human demonstration of the "shake" task in real-world scenario.

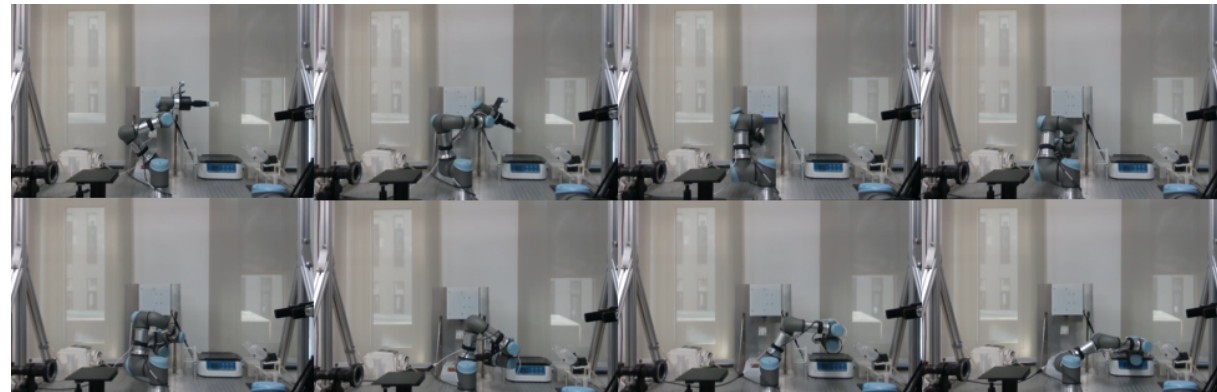

Figure 19: "Shake" task in real-world scenario.

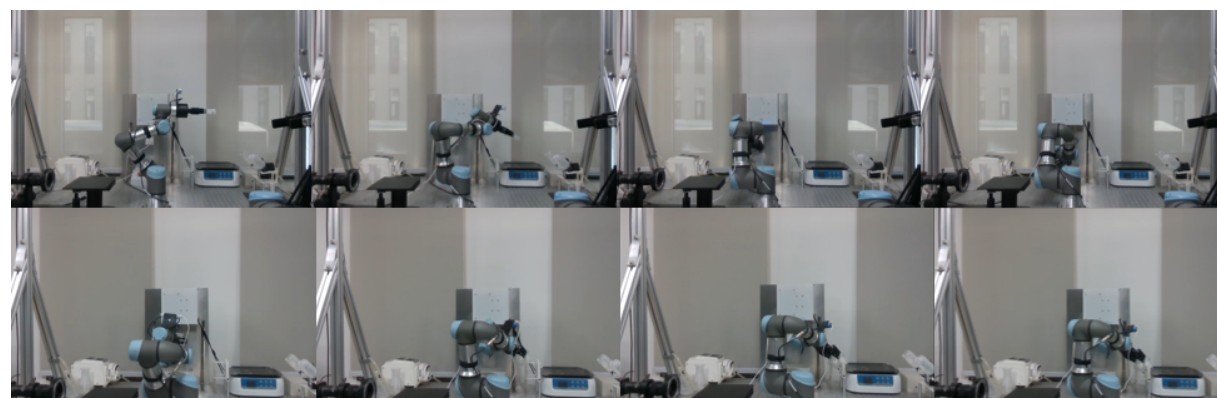

Figure 20: Failure case of the "shake" task in real-world scenario.

