# OpenReview forum: "Compositional Multimodal Reasoning for Long-Horizon Robotic Manipulation in Scientific Experiments"
_ICLR.cc/2026/Conference — ICLR 2026 Conference Withdrawn Submission_

### Official Review · Reviewer_7GdJ · 2025-10-19

**Soundness:** 3
**Presentation:** 2
**Contribution:** 2
**Rating:** 2
**Confidence:** 4

**Summary:**

The paper proposes CoMP, a modular framework for long-horizon robotic manipulation in scientific experiments. CoMP separates (i) a task-level LLM planner using chain-of-thought, (ii) a mid-level multimodal planner that predicts future scene frames and maps subtasks to action primitives, and (iii) a low-level RL controller (e.g., DDPG). The authors also introduce a simulation suite and benchmark, report gains over several baselines (including on RLBench), and present a limited sim-to-real evaluation using detection + depth for coordinate transfer.

**Strengths:**

- Clear modularity & interpretability: Decoupling planning, grounding, and control is well-motivated for long-horizon laboratory workflows and aids debugging/verification.
- New benchmark & analyses: A domain-specific benchmark emphasizing procedural dependencies is valuable.
- Empirical coverage: Results on both the proposed suite and RLBench show consistent improvements on several tasks

**Weaknesses:**

- Real-robot validation is limited: The physical experiments are narrow in scope and rely on a sim-to-real coordinate mapping; robustness (sensor noise, calibration drift, safety) is under-characterized. Strengthen with more tasks, seeds, and error/failure analyses; report success variance and safety incidents.
- Training–to–deployment gap: Clarify whether the RL policy is trained only in simulation and how it transfers (domain randomization, dynamics mismatch, contact modeling). Provide fine-tuning/none, and quantify performance drop from sim to real.
- Efficiency & latency: The pipeline appears complex (CoT planning → visual prediction → MLM primitives → RL). There is no analysis of runtime (per-step latency, FPS), computational cost, or throughput—critical for long-horizon tasks.
- Baseline completeness and fairness: Include comparisons to simpler Dual-System (System-1 + System-2) paradigms (e.g., “Hi Robot”-style simple planner + skills) and rule-based/task-graph planners to demonstrate that gains require the full CoMP stack. Explicitly discuss fairness for re-implemented baselines (training data, hyperparameters).

**Questions:**

- RL training & transfer: Was the RL policy trained exclusively in simulation? What domain randomizations were used? Any real-robot fine-tuning? How is stability ensured under perception errors?
- Calibration pipeline: How is the YOLO + depth → simulation mapping calibrated and maintained over time? What is the failure rate due to calibration drift or occlusions?
- Runtime profile: What are the end-to-end latencies for (a) CoT planning, (b) future-frame prediction, (c) MLM primitive generation, and (d) RL control? Where is the bottleneck?

---

> ### Author Response · Authors · 2025-11-19
> **Response to Reviewer 7GdJ (1)**
>
> We thank the reviewer for the constructive feedback. We respond to each point below.
>
> **Q1. Limited real-robot validation**
>
> **R1:** We agree that sim-to-real robustness is essential. Our approach is actually more interpretable and easier to debug than end-to-end VLA transfer, because failures can be traced to specific sensing or calibration issues rather than opaque model behavior.
> Sensor noise is the dominant error source, and planning and control remain stable after transfer. We have added more real-world tasks and trials (the following table) and will include analysis of both successes and failures in the revision.
>
> |   Task   |                Description                | Training data | Success rate |
> |:--------:|:-----------------------------------------:|:-------------:|:-------------:|
> |  Weigh   | Place containers on the balance for automated weighing. |      27      |     66.7%     |
> |  Shake   | Load the container onto the shaker.       |      25      |     70.0%     |
>
> **Q2. Training-to-deployment gap**
>
> **R2:**
> The RL controller is trained entirely in simulation, and for the UR3e platform we use, this does not introduce a meaningful dynamics-level gap. The observed performance drop comes almost exclusively from perception noise (e.g., depth errors), rather than control or dynamics mismatch.
> These failures can be mitigated through higher-precision sensors or redundant perception setups, rather than changes to the controller itself.
> We include a brief analysis of these real-robot discrepancies in the revision.
>
> **Q3. Efficiency \& latency**
>
> **R3:**
> As for efficiency, the system does not introduce additional runtime overhead.
> Because the architecture is decoupled, computation-intensive reasoning is performed once before execution, while the low-level controller runs with millisecond-level latency during closed-loop control. The following table provides detailed per-module inference times and specifies which components run online versus offline. Overall, the modular structure improves robustness and interpretability while preserving real-time performance.
> The following shows the inference time of each module.
>
> | Component       | Inference latency (ms) | Runs Online? |
> |:---------------:|:----------------------:|:------------:|
> | LLM             | 3024                   | No           |
> | MP              | 5714                   | Optional     |
> | MLM             | 3441                   | Optional     |
> | RL              | 15                     | Yes          |
> | **Online total** | **15**                 | -            |
>
> **Q4. Baseline completeness and fairness**
>
> **R4:**
> CoT-VLA already represents such a System-1 + System-2 design, and our experiments show that it underperforms under the same data conditions. For fairness, all VLA baselines are fine-tuned using exactly the same PerAct training data, with default hyper-parameters.
> (For OpenVLA, we follow the official ``pi0\_liberolow\_mem\_finetune`` recipe due to GPU memory constraints, ensuring a faithful and comparable fine-tuning setup.)
> Our diffusion module, in contrast, did not use trajectory supervision.
>
> As for Hi-Robot, it combines GPT-4o planning + OpenVLA execution, whose performance could be close to our “MLM + LR” ablation in Table 3 of the original submission.
> The performance gap mainly reflects the generalization of OpenVLA, not planning quality. We will include the comparison when we conduct the reimplementation.
>
> **Q5. RL training \& transfer**
>
> **R5:**
> The RL controller is trained entirely in simulation with no real-robot fine-tuning in our work.
> Because the UR3e arm has very accurate and well-modeled kinematics, we observe almost no control-level sim-to-real gap; the performance drop on hardware comes almost entirely from perception noise rather than dynamics mismatch. Since the low-level controller operates in closed-loop, stability under perception error is largely maintained, and additional robustness can be achieved simply by improving sensor quality rather than re-training the policy.
>
> **Q6. Calibration pipeline**
>
> **R6:**
> The potential sources of calibration error primarily stem from camera lens distortion and sensor noise, both of which are dependent on the relative position between the camera and objects. These errors can be mitigated using multiple camera viewpoints if needed, but are beyond the scope of our work. Given that our camera views are fixed (unlike first-person views often used in VLA) and occlusions are minimal, the impact of calibration drift is negligible in our setup.
>
> **Q7. Runtime profile**
>
> **R7:**
> Please refer to the response to Q3.

---

> ### Author Response · Authors · 2025-11-19
> **Response to Reviewer 7GdJ (2)**
>
> **Q5. Scalability concerns**
>
> **R5:**
> Our design does not introduce unnecessary engineering overhead; rather, it reduces data dependence and facilitates transfer in laboratory environments.
> Lab tasks exhibit high visual variability, precise physical constraints, and strong domain shifts.
> End-to-end policies or IL pipelines typically require paired robot-action demonstrations, which are extremely costly or infeasible in such settings.
> In contrast, our framework decomposes the problem into modular components operating at different levels. High-level semantic reasoning leverages existing knowledge, while mid- and low-level modules handle visual grounding and control, minimizing the need for expensive data collection.
> Collecting raw visual demonstrations or simple bounding-box annotations is low-cost, does not require expert operators, and is easily standardized, making the approach highly scalable.
> Adding new tasks or objects does not increase model complexity, but only high-level prompts or minor fine-tuning of the low-level modules are needed.
> In this sense, the multiple components are not an overhead but the key to maintaining data efficiency, robustness, and scalability in challenging lab scenarios where monolithic end-to-end methods often fail.
>
> **Q6. Dual decomposition**
>
> **R6:**
> Please refer to the response to Q1.
>
>
> **Q7. Novelty and positioning**
>
> **R7:**
> Please refer to the response to Q2.
>
> **Q8. Imitation learning**
>
> **R8:**
> Regarding the request for confidence intervals and more detailed physical-robot statistics, in addition to the clarification provided in Q3 and Q4, we would like to note that most prior VLA papers do not report confidence intervals for full embodied tasks.
> Unlike conventional perception or recognition benchmarks, an embodied perception–planning–control pipeline contains multiple stochastic factors (scene initialization, object configuration, language variation, controller noise, etc.). Searching for “lucky seeds’’ is almost impossible, and the evaluation noise is dominated by underlying scene randomness rather than model stochasticity.
>
> For our benchmark, we therefore adopt a large-scale randomized evaluation instead of repeated trials on fixed scenes. For each task, we randomly pair diverse GPT-generated instructions with multiple objects to form 20 different input prompts, and for each combination, we sample 50 randomized scenes with different object locations, yielding 1,000 inference trials per task.
> This setting makes the notion of a standard confidence interval ill-defined, since the variance primarily comes from intentionally diversified scene distributions rather than repeated measurements of a fixed setup.
>
> **Q9. Real-world validation**
>
> **R9:**
> Please refer to the response to Q4.
>
>
> **Q10. Scalability and cost**
>
> **R10:**
> Most components of our framework incur negligible training cost.
> All LLM/MLM modules are used off-the-shelf without any fine-tuning.
> The only modules that require training are the diffusion-based multimodal predictor and the lightweight RL controller.
> Among them, the diffusion model is the most computationally expensive, while the RL policy can be trained even on a single PC-level GPU (e.g., RTX 4060, 70ms). The following table reports per-module inference latency; except for the RL controller, all reasoning stages can be completed entirely before task execution.
> To avoid potential error accumulation, both the diffusion model and the MLM can be reconditioned with real-time observations during execution if desired (though we do not enable this in the current paper).
>
> | Component       | Inference latency (ms) | Runs Online? |
> |:---------------:|:----------------------:|:------------:|
> | LLM             | 3024                   | No           |
> | MP              | 5714                   | Optional     |
> | MLM             | 3441                   | Optional     |
> | RL              | 15                     | Yes          |
> | **Online total** | **15**                 | -            |

---

### Official Review · Reviewer_dB6F · 2025-10-31

**Soundness:** 2
**Presentation:** 3
**Contribution:** 2
**Rating:** 2
**Confidence:** 4

**Summary:**

The paper proposes CoMP, a modular framework for long-horizon robotic manipulation in scientific lab settings. It decouples: (i) a task-level LLM planner using chain-of-thought (CoT) with a verification–correction loop, (ii) a mid-level multimodal planner combining a conditional diffusion–based future frame predictor and a multimodal LLM to turn predicted goals into action primitives, and (iii) a low-level RL controller (DDPG). The authors also introduce a small simulation benchmark in CoppeliaSim and report higher success rates than several baselines on their benchmark and mixed results on RLBench, plus limited sim-to-real tests using YOLOv8 + depth mapping.

**Strengths:**

1. The modular decomposition is easy to understand: a language-based task planner decomposes goals into symbolic subtasks, a mid-level component grounds each subtask using the current and predicted goal images, and a low-level controller executes the resulting action primitives. The paper’s overview figure and method section make this pipeline intuitive.
2. Clear ablations illustrate the role of key modules: the paper varies the LLM and the MLM and reports CoT step-wise ablations, which together help attribute where gains come from.
3. The descriptions of each module are clear in both the main text and the appendix, with concrete prompt templates, training details, and a breakdown of the verification–correction operations; this level of disclosure makes the system easier to understand and (partially) reproduce.

**Weaknesses:**

1. **Overly complicated pipeline.** The system performs subtask decomposition twice—first via a task-level LLM planner and then again via GPT-4o in the mid-level. It’s unclear why a single VLM could not directly produce the decomposition and grounding, and what unique value the initial LLM adds.

2. **Limited novelty.** The core ideas—LLM/VLM-based subtask decomposition and future image prediction—are well-trodden areas with substantial prior art. The paper does not sufficiently articulate what is new beyond combining known components.

3. **Missing imitation-learning setting.** Most competitive VLA systems and many baselines in the literature rely on imitation learning. The paper uses only RL, making comparisons incomplete and potentially unfavorable to the method’s practicality.

4. **Incomplete empirical validation.** Experiments are almost entirely in simulation with only minimal real-robot results. Claims about long-horizon manipulation and “scientific workflow” utility are not substantiated on physical hardware.

5. **Scalability concerns.** The many moving parts (planner LLM, GPT-4o grounding, predictor, RL controller) raise questions about data efficiency, engineering overhead, and whether the framework scales to large, diverse task suites without prohibitive complexity.

**Questions:**

1. **On the dual decomposition:** Why is a separate task-level LLM needed if a modern VLM can directly output grounded subgoals/action primitives?

2. **On novelty and positioning:** What specific technical contributions differentiate this work from prior LLM/VLM planning and future-prediction pipelines? Beyond integration, are there new algorithms, training objectives, or guarantees? A related-work table mapping differences would help.

3. **On imitation learning:** Can you include IL baselines and/or an IL variant of your method? If not, please justify why RL is necessary here and report how many demonstrations or episodes would be needed to match RL performance.

4. **On real-world validation:** Can you expand physical-robot evaluation (more tasks, repetitions, failure breakdowns) and report success metrics with confidence intervals? What are the dominant real-world failure modes (perception vs. planning vs. control)?

5. **On scalability and cost:** What is the training/inference cost and data requirement of each module, and how do these scale with task count and horizon length? Any evidence that the architecture remains tractable and reliable when moving to a large multi-task setting?

---

> ### Author Response · Authors · 2025-11-19
> **Response to Reviewer dB6F**
>
> We thank the reviewer for the constructive feedback. We respond to each point below.
>
> **Q1. Overly complicated pipeline**
>
> **R1:**
> We use a hierarchical design not for architectural complexity, but because scientific-experiment tasks suffer from (i) lack of paired demonstrations, (ii) large domain gaps from common VLM training data, and (iii) high-level procedural abstraction requirements. A single end-to-end VLM cannot reliably bridge these gaps.
>
> In fact, as shown in Table 3 in the original submission, we replaced the first two modules with either LLaMA3.2 or GPT-4o, keeping prompts the same as our method.
> Results show substantial performance drops on high-level tasks (e.g., ``crystallize``). Even with strong long-context reasoning, one-shot long prompts yield uncontrolled multi-step plans, whereas our modular hierarchy enforces structure and improves reliability.
>
> **Q2. Limited novelty**
>
> **R2:**
> While LLM/VLM-based decomposition and visual prediction exist, existing methods struggle in scientific lab settings that are OOD to general scenarios.
> There are problems with high visual complexity, rare or safety-critical objects, and limited paired vision–action data that make end-to-end training inefficient or unstable.
> Our framework addresses these issues by decoupling high-level semantic reasoning from low-level control and leveraging simulation and lightweight visual grounding.
> Although the pipeline has multiple components, where each is necessary, introducing new algorithms would confound validation.
> Our main novelty lies in problem modeling and principled decoupling, not in designing new low-level models where existing methods suffice.
>
>
>
> **Q3. Missing imitation-learning setting**
>
> **R3:**
> While most VLA baselines rely on imitation learning (IL), collecting paired video–action data in scientific labs is costly and often impractical due to consumable materials and safety constraints.
> In contrast, our RL-based controller benefits from the decoupled framework: high-level semantics are learned from flexible demonstrations (manual or simulated), and low-level execution is handled by RL without requiring trajectory supervision. This enables effective learning in real lab scenarios where standard IL would struggle.
>
> **Q4. Incomplete empirical validation**
>
> **R4:**
> Our work focuses on effectively transferring existing large models to laboratory scenarios. Our benchmark is specifically designed to reveal the limitations of directly fine-tuning VLAs under visual OOD conditions and complex procedural dependencies.
> To further validate our approach, we added two real-world tasks difficult to model in simulation, executed purely in the real environment with human-performed demonstrations as training data.
> Preliminary results are shown in the following table, and full experiments and visualizations will be included in the revision.
> |   Task   |                Description                | Training data | Success rate |
> |:--------:|:-----------------------------------------:|:-------------:|:-------------:|
> |  Weigh   | Place containers on the balance for automated weighing. |      27      |     66.7%     |
> |  Shake   | Load the container onto the shaker.       |      25      |     70.0%     |
>
> The simulation environment is primarily provided for reproducibility, allowing readers to separately examine software and hardware effects.
> Regarding sim-to-real transfer, performance degradation is mainly due to visual perception.
> Low-level control accuracy is largely preserved.
> Our method allows correcting control errors via light RL fine-tuning or explicit error-compensation modules.
> In contrast, VLA approaches rely on end-to-end training to implicitly handle low-level errors and may require retraining if the operational environment changes.
> Our modular approach obviously provides greater flexibility and robustness in real lab scenarios.

---

### Official Review · Reviewer_LRa6 · 2025-11-01

**Soundness:** 2
**Presentation:** 3
**Contribution:** 3
**Rating:** 4
**Confidence:** 3

**Summary:**

This paper proposes a non end-to-end pipeline for long-horizon robotic experimentation.

Specifically, the framework first uses a large language model (LLM) to decompose a high-level experimental goal into a sequence of symbolic subtasks {s}, then employs a multimodal language model (MLM) to further break down each subtask into fine-grained action primitives. Finally, a reinforcement learning (RL) controller is used to train and execute the corresponding physical actions.

The paper also introduces a simple benchmark and training dataset containing basic chemical operations (e.g., pouring, stirring, mixing).
Experimental results show that the proposed pipeline, CoMP, achieves state-of-the-art performance on both the proposed benchmark and public datasets.

**Strengths:**

* Clear motivation: the authors convincingly argue that end-to-end approaches fail to generalize in long-horizon scenarios (due to catastrophic forgetting), and therefore adopt a hierarchical decomposition strategy (LLM -> MLM -> RL). The logic of this design is coherent and well-motivated.

* Dataset contribution: the paper provides a compact but useful benchmark simulating chemical laboratory operations.

* Strong results: CoMP achieves SOTA performance across evaluated benchmarks.

**Weaknesses:**

* In the public benchmark (Table 4), CoMP performs poorly on the "pick cup" task. This is surprising since a long-horizon planner should, in theory, handle short-horizon tasks more easily, i.e., it shouldn't hurt the original capability of short-horizon tasks.

* The dataset is overly simple, covering only a limited set of basic chemical actions.

* It is unclear how fine-grained CoMP’s control actually is, i.e., in the “pour” task, does the system control liquid volume (e.g., in milliliters), or is the simulation limited to a symbolic pouring gesture without real fluid modeling?

* The RL training procedure is insufficiently described: is the controller trained from scratch, or initialized via imitation learning?

**Questions:**

The questions are included in the weaknesses.

---

> ### Author Response · Authors · 2025-11-19
> **Response to Reviewer LRa6 (1)**
>
> We thank the reviewer for the constructive feedback. We address the key concerns below.
>
> **Q1. Short-horizon task performance**
>
> **R1:** We appreciate the reviewer’s observation.
> The lower performance on `Pick Cup` is not due to limitations of the long-horizon design, but rather from a specific constraint we imposed in our real-lab setting.
> For safety reasons (i.e., to avoid crushing fragile glassware or causing slippage with wide-mouth vessels), we enforce a side-wall grasp strategy rather than the standard top-down or full-envelope grasp used by most baselines.
> This grasping strategy is appropriate for laboratory containers but turns out to be suboptimal for the cups used in the public benchmark, where the geometry makes side-grasping less stable and more prone to slippage.
>
> We thank the reviewer for pointing this out. We will include results with the safety constraint relaxed.
>
> **Q2. Dataset simplicity**
>
> **R2:** We respectfully disagree with the claim that our dataset is “overly simple.”
> Our dataset focuses on the most common and fundamental operations in real laboratory workflows, and it already covers the majority of fundamental experimental primitives as well as two longer, multi-step procedures (and two additional long-horizon tasks in the revision).
> Importantly, many public robotic datasets contain fewer primitive categories, often dominated by variations of pick-and-place with different objects.
>
> To further stress-test generality, we additionally introduced two new tasks with higher difficulty, where the mid-level planner is trained only on human-performed demonstrations and where the simulator does not contain the new-task scenes.
> Preliminary results (the following table) show that CoMP still performs reliably under this stronger domain shift. We will include the full experimental results in the revision.
>
> |   Task   |                Description                | Training data | Success rate |
> |:--------:|:-----------------------------------------:|:-------------:|:-------------:|
> |  Weigh   | Place containers on the balance for automated weighing. |      27      |     66.7%     |
> |  Shake   | Load the container onto the shaker.       |      25      |     70.0%     |

---

> ### Author Response · Authors · 2025-11-19
> **Response to Reviewer LRa6 (2)**
>
> **Q3. Fine-grained control**
>
> **R3:** Our current simulation models liquid using particle-based approximations. Thus, the system does not target milliliter-level volumetric control.
> Importantly, this aspect is orthogonal to the proposed CoMP framework.
> Precise fluid regulation can be integrated as an additional feedback module without altering the hierarchical planning or grounding pipeline.
> We thus exclude this component to focus on evaluating the modular reasoning and manipulation capabilities, but CoMP is compatible with any future fine-grained fluid-control module.
>
> **Q4. RL training procedure**
>
> **R4:** Thanks to our decoupled design, the RL controller converges easily without imitation learning initialization.

---

### Official Review · Reviewer_wT92 · 2025-11-03

**Soundness:** 2
**Presentation:** 3
**Contribution:** 2
**Rating:** 4
**Confidence:** 3

**Summary:**

- This paper presents CoMP, a compositional and decoupled framework for long-horizon robotic planning in scientific experiments.

- CoMP combines task-level CoT decomposition, multimodal visual prediction, and RL-based control for robotic planning and control.

- This paper also introduces a benchmark dataset for scientific experiment tasks.

**Strengths:**

- It is sound to employ a modular approach rather than an end-to-end method for long-horizon robotic manipulation in laboratory environments.

- Focusing on autonomous experimental systems for science is highly insightful.

**Weaknesses:**

- A primary concern is the potential for the entire system to be overly complex, redundant, and time-consuming. Why did the authors choose not to leverage a single powerful Vision-Language Model (VLM), such as GPT-4o or Gemini, to handle both task-level planning and mid-level planning concurrently?

- Is the visual prediction module essential for planning? For a fairer comparison in Table 3, comparing MLM (LLaMA3.2)+RL and MLM+RL is insufficient due to the different base MLLMs used. The authors should have conducted an ablation using MLM (GPT-4o, without vision input) + RL as a baseline. Furthermore, I argue that visual prediction may not be strictly necessary for sub-task planning, as similar grounding could potentially be achieved by employing highly detailed text prompts.

- The visual prediction module necessitates the use of expert demonstration samples. Therefore, the authors' claim of operating "without trajectory-level supervision" is arguably misleading. Although action information is not explicitly used, the cost of acquiring the required demonstration data is comparable to that of standard Imitation Learning (IL) methods.

- The comparison against several IL based works may not be entirely fair, as those baselines do not require a separate visual prediction module. What would be the performance outcome if the authors trained the IL policy models using the exact same demonstration data?

**Questions:**

Please see the Weaknesses section.

---

> ### Author Response · Authors · 2025-11-19
> **Response to Reviewer wT92 (1)**
>
> We thank the reviewer for the detailed and constructive feedback. Below we address the key concerns point by point.
>
> **Q1. System complexity**
>
> **R1:**  We use a hierarchical design not for architectural complexity, but because scientific-experiment tasks suffer from (i) lack of paired demonstrations, (ii) large domain gaps from common VLM training data, and (iii) high-level procedural abstraction requirements. A single end-to-end VLM cannot reliably bridge these gaps.
>
> In fact, as shown in Table 3 in the original submission, we replaced the first two modules with either LLaMA3.2 or GPT-4o, keeping prompts the same to our method. Results show substantial performance drops on high-level tasks (e.g., `crystallize`). Even with strong long-context reasoning, one-shot long prompts yield uncontrolled multi-step plans, whereas our modular hierarchy enforces structure and improves reliability.
>
> As for efficiency, the system does not introduce additional runtime overhead.
> Because the architecture is decoupled, computation-intensive reasoning is performed once before execution, while the low-level controller runs with millisecond-level latency during manipulation execution. The following table provides detailed per-module inference times and specifies which components run online versus offline. Overall, the modular structure improves robustness and interpretability while preserving real-time performance.
> MP and MLM also support online updates.
> After each subtask execution, they can update the remaining actions to reduce error accumulation. In this work, we present the extreme case where all high-level reasoning occurs before execution. Considering that subtasks are executed within seconds, we recommend reducing MP input/output resolution to 128×128 for online use.
>
> | Component       | Inference latency (ms) | Runs Online |
> |:---------------:|:----------------------:|:------------:|
> | LLM             | 3024                   | No           |
> | MP              | 5714                   | Optional     |
> | MLM             | 3441                   | Optional     |
> | RL              | 15                     | Yes          |
> | **Online total** | **15**                 | -            |
>
> **Q2. Necessity of the visual prediction module**
>
> **R2:** We have conducted the suggested comparison using MLM (GPT-4o without visual input) + RL, and the results are reported in the following table. As expected, removing the visual input significantly degrades performance.
> The model fails to identify specific objects and their states, making reliable sub-task execution impossible. This confirms that the visual prediction module is indeed important for grounding actions and that highly detailed text prompts alone are insufficient to replace it in our laboratory manipulation setting.
>
> |      Task               | Pick_place | Pour | Stir | Mix  | Crystallize |
> |:-----------------------:|:----------:|:----:|:----:|:----:|:-----------:|
> | CoMP w/o visual input   |   70.4     | 54.6 | 34.8 | 23.8 |    12.4     |
> | CoMP w/o MP (MLM+RL)    |   78.9     | 62.6 | 70.5 | 43.2 |    21.0     |
> | CoMP                    |   80.4     | 64.6 | 74.4 | 52.2 |    30.2     |

---

> ### Author Response · Authors · 2025-11-19
> **Response to Reviewer wT92 (2)**
>
> **Q3. Training data of the visual prediction module**
>
> **R3:** While we do rely on expert demonstration videos to train the multimodal predictor (MP), these demonstrations are different from IL datasets in both cost structure and data-collection constraints, especially in laboratory environments.
> In generally used household manipulation domains, collecting paired (video, action trajectory) sequences is relatively easy and inexpensive.
> In real scientific labs, however, demonstrations are performed by trained experts, materials are consumable, and repeatedly executing full workflows with a robot arm is somewhat impractical.
> Our framework leverages the VLM’s semantic extraction to enable semantic-level transfer.
> High-level demonstrations can be captured much more flexibly, such as manual operation demonstrations and simulation-only renderings, without requiring synchronized robot trajectories, while the low-level execution is handled by a separate controller.
>
> To further show this flexibility, we additionally evaluate scenarios where some laboratory tools are difficult to model in simulation by adding 2 new tasks.
> We therefore incorporated real human-executed experiment videos as demonstrations and observed that our method can still handle these cases effectively.
> Preliminary results are shown in the following table, and full experiments will be included in the revision.
>
> |   Task   |                Description                | Training data | Success rate |
> |:--------:|:-----------------------------------------:|:-------------:|:-------------:|
> |  Weigh   | Place containers on the balance for automated weighing. |      27      |     66.7%     |
> |  Shake   | Load the container onto the shaker.       |      25      |     70.0%     |
>
> **Q4. Comparison with IL models**
>
> **R4:** We apologize for the lack of clarity in the original submission.
> In Table 4 of the original submission, the IL baselines are fine-tuned using exactly the same demonstration data as our system.
> All VLAs are fine-tuned following the protocol used in PerAct, and the training data are the same to our MP, except that our MP only uses visual observations, without the paired action trajectories.
>
> In this sense, if anything, the comparison is conservative against our method.
> The IL baselines receive strictly more supervision (full vision–trajectory pairs), while our model is trained without any trajectory-level labels.
> Despite this stronger supervision, their performance remains lower than ours.
> A likely explanation is that straightforward fine-tuning of VLA policies struggles to cope with domain shift (e.g., viewpoint or object placement variations), whereas our decoupled architecture isolates high-level semantics from low-level control and thus transfers more robustly.
>
> We will make this experimental setup clearer in the revision.

---

### Author Response · Authors · 2025-11-19
**General Response**

# General Response
We thank all reviewers for their constructive feedback. We first address a common concern regarding the rationality, necessity, and novelty of our design and benchmark.

Overall, our system is **not an overly complex or ad-hoc design**. It directly targets the challenges inherent in scientific experimental environments, where existing household-trained VLA models fail to generalize.

Our framework uses a **heterogeneous decoupling** strategy, where each module handles only the information it strictly needs.
This is not only theoretically grounded but also empirically validated by the distribution shift introduced in our benchmark.

## Benchmark motivation
Our benchmark collects a class of scientific manipulation tasks whose **visual**, **concept domain**, and **task-logic** characteristics differ from those commonly found in household or tabletop settings. Only some low-level physical relations (e.g., containment or support) are the same as standard manipulation tasks.

These differences make it difficult to reuse existing paired vision–trajectory datasets, and collecting new demonstrations of robot manipulations in real labs is costly and safety-sensitive. We therefore observe that directly applying or lightly fine-tuning SOTA VLAs leads to degraded performance, suggesting that the distribution shift is non-trivial.

## Learning and sim-to-real transfer
Instead of recollecting an expensive domain-specific multimodal dataset, our pipeline allows each component to be trained with the **cheapest available supervision**.

This eliminates the need for vision–action demonstrations while outperforming fine-tuned VLAs.
Moreover, our method makes sim-to-real transfer practical: only the visual grounding module (e.g., YOLO) requires minor adaptation.
The controller requires little or no additional fine-tuning due to the stable lab environment.

## Cognition and information theory
Our design aligns with the Dual-System principle, as mentioned by 7GdJ.
The task-level planner provides System-2 task-level reasoning, while the remaining two modules serve as System-1 perception and action.Recent works, e.g., Hi Robot (ICML 2025) and CoT-VLA (CVPR 2025), highlight using this paradigm as their novelty, and our contribution lies in applying it to scientific experimentation and offering a principled decoupling implementation.

Also, lab visuals are high-entropy and noise-prone, while action trajectories are low-entropy and physically constrained. End-to-end learning may cause severe mutual-information imbalance. Our three-level structure restores information efficiency and stabilizes learning.

## About the revision submission

A revison has be summited.

---

### Note · Authors · 2026-01-19

I have read and agree with the venue's withdrawal policy on behalf of myself and my co-authors.